# Solar Distillation of Impure Water from Four Different Water Sources under South-Western Nigeria Climate

Saheed A. Adio[1,], Emmanuel A. Osowade[1], Adam O. Muritala[1], Adebayo A. Fadairo[1], Kamar T. Oladepo[2], Surajudeen O. Obayopo[1] and P. Fase[1]

[1]Thermofluids Research Group, Department of Mechanical Engineering, Obafemi Awolowo University, Ile-Ife.
[2]Water Engineering Research Laboratory, Civil Engineering Building, Department of Civil Engineering, Obafemi Awolowo University, Ile-Ife.

*Correspondence to*: Adam Olatunji Muritala (muriadam@gmail.com)

**Abstract.** The enormous problems caused by scarcity of potable water and transmission of water-borne diseases such as Cholera, Dracunculiasis, Hepatitis, Typhoid and Filariasis in some parts of Nigeria have created a public health concern. Thousands of lives are wasted daily due to contact with water-borne diseases. The insufficient medical resources available in developing countries are deployed towards the treatment of water-borne diseases that can easily be avoided if potable water can be made available. This study seeks to investigate purification of four different water samples (namely, water from flowing river, freshly dug well or groundwater, rainwater from the rooftop, and heavily polluted dirty water) consumed by the people in the local community using solar desalination method. A single basin solar still was constructed and experimental studies were carried out to determine the influence of solar insolation and temperature variations on the yield of the distillate for both passive and active solar still tested. The quality of the distillate was tested by measuring the total dissolved solid (TDS) and electrical conductivity (EC) and later compared to World Health Organization (WHO) standard for drinkable water. The values obtained after desalination falls within the acceptable/tolerable range for TDS and EC in accordance with the World Health Organization standard for quality drinkable water. This analysis provides an indigenous distillation method to enhance production of drinkable water at low cost.

## 1 Introduction

Water is a major resource most living and non-living organisms depend on. It plays key roles in the sustenance of life, economic and general well-being of a nation. It is one of the most abundant resources on earth, covering three-fourths of the planet's surface. About 97% of the earth's water is present as salt water in oceans and the remaining 3% as fresh water in the form of ice, groundwater, lakes, and rivers. Less than 1% fresh water is within human reach (Manokar et al., 2014). Naturally, most water exists in a polluted or non-purified form with lots of microorganisms capable of causing Cholera, Dracunculiasis, Hepatitis, Typhoid, Filariasis and so on (Rab, M. A., Bile, M. K., Mubarik, M. M., Asghar, H., Sami, Z., Siddiqi, S., ... & Burney, M. I., 1997). In the world, 3.575 million people die each year from water-related diseases (Adeyinka, S. Y., Wasiu, J., & Akintayo, O. C., 2014) and 1.1 billion people out of the world's population lack access save drinkable water in 2017, 785 million people still lack a basic water service and among them 144 million people still collected drinking water directly from rivers, lakes and other surface water sources (World Health Organization (WHO), 2002). Potable water scarcity is a growing problem for large regions in the world and the primary drivers are proliferating world population growth, industrialization and

urbanization, irrigation in agriculture and the higher consumption rate associated with rising standards of living. Also, existing water resources are expected to be affected by the global climate change, thereby altering the distribution of wet and arid regions and raising the salinity of some coastal aquifers (Summers et al., 2012). These factors with the inherent deadly water-borne diseases accompanying impure water usage are pointers to the urgent need for the purification of water that is otherwise too saline for human consumption. Most times, the water consumed by people in sub-Sahara Africa, Nigeria specifically is sourced from a flowing river, freshly dug well or groundwater, rainwater from rusted rooftops and heavily polluted water. Many water purification processes exist including desalination technology. There are over 10,000 desalination plants in the world, with a total desalted water capacity of over 5 billion gallons a day. Saudi Arabia is the largest user of desalination with about 25 percent of the world capacity, and the United States is the second largest user with 10 percent (Cengel, Yunus A. and Michael A. Boles., 2002). Vapour compression distillation, reverse osmosis and electrodialysis using electricity generated from coal and fossil fuels combustion as input energy are examples of desalination systems, however, they have been found to be very expensive and unsustainable basically due to the amount and cost of energy required to carry out the processes. Also, the hazardous greenhouse gases emission released during desalination processes using electricity from fossil fuel combustion causes climate change and ozone layer depletion which in turn results in rise in global temperature and melting of glaciers and ice sheets faced by many countries of the (Goosen, M., Mahmoudi, H., & Ghaffour, N., 2012; Kalogirou, S. A., 2013; Kalogirou, 1985). Currently, solar desalination stands as one of the most efficient, effective and more economical in terms of low running cost, long lifespan and low or no environmental pollution when compared with other types of water purification systems most especially for rural communities. This can be attributed to the free and abundant gift of the sun and its renewability (Elango et al., 2015a; Sampathkumar et al., 2010). The device used for performing this purpose is solar still. It operates similarly to the natural hydrologic cycle of evaporation and condensation. Among the different types of solar stills, single basin single slope occupied the best place due to its simplicity in design and operation. The heat from the sun evaporates the pure water from the impure, brackish or saline water collected in the still basin covered by a glass leaving behind the microorganisms and other contaminants in the basin. The evaporated water condenses on the inner surface of the glass, the condensed liquid flows down freely beneath the inclined cover to a V-shape trough/water channel at the bottom of the still where it is collected for human consumption (Tiwari, A. K., & Tiwari, G. N., 2006; Tiwari et al., 2009).

Many settlements are facing problems caused by potable water scarcity in Nigeria daily. They result to drinking water sourced from flowing river, freshly dug well and rainwater falling off rooftops for cooking and drinking during the raining season without any further purification (Onwujekwe et al., 2009; Smith et al., 2004). The drinking of water from these sources without further purification poses health challenges to different rural settlements in this category. For instance, most rooftops are rusted iron sheets and rainwater collected from these rooftops are not only dirty but may be carcinogenic (Abbasi and Abbasi, 2011; Bennamoun et al., 2013; González, 2012; Lye, 2009; Meera and Ahammed, 2006; Mendez et al., 2010; Norman et al., 2019; University of Texas at Austin, n.d.).

Apart from the coastal region of Nigeria where people are forced by circumstance to process salty water for domestic use, the commonly available water in some rural areas is not pure due to dissolved organic and inorganic materials. In some locations (e.g Ile-ife Osun state: 7.4905°N, 4.5521°E) the salt content of water fetched from dug wells, rivers and even bore hole is very high and requires treatment. Hence, an affordable, yet very efficient process

that requires little or no technical know-how and maintenance for the purification of water from these sources will
be a welcome idea in such rural settlement. Therefore, the main objective of this study is to design, construct and
test a solar desalinating plant made with locally sourced materials for the purification of water from the following
sources; rainwater, freshly dug well water, river water and heavily polluted water which are peculiar to the site
where this research is carried out and most rural areas in Nigeria.
The effects of the solar radiation intensity, inner glass surface temperature and the absorber plate temperature as
they affect the hourly distillate yield was examined for both passive and active solar still configurations. The
performance and efficiency of the solar desalinating plant was evaluated based on its distillate yield. Finally, the
water samples were tested after desalination to ascertain suitability of the water for drinking purpose based on the
WHO standard for drinking water. This is with a view to mitigate the widespread of water-borne diseases in rural
settlements in Nigeria as a result of indiscriminate drinking of untreated or impure water due to unavailability of
drinkable water.
**2 Literature survey**
Solar still can be classified into two; active and passive solar still. Passive solar still receives solar radiation directly
from the sun into the water in the basin. It is the only source of energy responsible for raising the water temperature
for evaporation. Active still utilizes more than one energy source other than the sun for water distillation (El-Sebaii,
A. A., 2004; Sivakumar and Sundaram, 2013). The extra thermal energy is supplied through an external means for
better performance. The temperature difference between the water in the basin and the inner surface of the glass
cover, the water depth in the basin, material of the basin and the black body absorber, wind velocity, insolation
intensity, ambient temperature and inclination angle of the glass have been found to affect the solar still productivity
(Elango et al., 2015b; Sampathkumar et al., 2010; Tiwari, A. K., & Tiwari, G. N., 2005). Although solar distillation
is not a new technology, likewise the method/structure of solar still (that is single slope conventional type) adopted
in this research. However, the experimental design and the setup are location specific. This determines the angle of
tilts (that is the orientation and placement) of the solar still for better capturing of the solar radiation from the sun.
The tilt angle of the glass condenser significantly affects the output of the solar still. Many authors have worked on
the choice of optimum tilt angle for the glass cover. Amongst other, (Chinnery, 1971; Elsayed, 1989; Felske, 1978;
Heywood, 1971; Khorasanizadeh et al., 2014; Qiu and Riffat, 2003; Stanciu and Stanciu, 2014) obtained latitude +
$10^o$ tilt angle for better solar still performance. In the case of this study carried out on 7.5175° N latitude, the glass
cover tilt angle was kept at 17°52'', (that is 7.5175° N latitude plus 10°).
The performance of SS (that is the rate of evaporation of the impure water) is usually expressed as the amount of
distilled water produced by basin area in a day (Kabeel et al., 2014a). This performance is strongly enhanced by the
large temperature difference between the surface of the water in the basin (serving as the evaporator) and inner glass
cover surface (serving as the condenser) (Asbik et al., 2016; Elango et al., 2015a, 2015b; Kabeel et al., 2014b, 2016;
Manokar et al., 2014; Rahbar et al., 2015; Sampathkumar et al., 2010; Sharshir et al., 2016; Sivakumar and
Sundaram, 2013; Taghvaei et al., 2015). This quantity produced varies largely with the available solar radiation,
cloud conditions, atmospheric humidity, wind speed and ambient temperature, which are meteorological parameters
that cannot be altered by human beings. Other design parameters that affect productivity are the orientation of the
still, depth of water, inclination of the glass cover, slopes of the cover, insulation materials, area of absorber plate,

the inlet temperature of water and the temperature difference between the glass cover and the basin water (Sivakumar and Sundaram, 2013). This research compares the effect of passive solar still against the active type based on their respective distillate yield. The efficiency of the SS was evaluated based on its hourly distillate productivity rate. Also, the distillate (SS output) was analyzed based on the Electrical Conductivity (EC) and Total Dissolved Solid (TDS).

The salinity of any water strongly depends on the electrical conductivity and the TDS of the water. The TDS was measured to know the amount of both the organic and the inorganic materials that are dissolved in the water. The electrical conductivity was also measured to know how well the desalinated water can conduct electric current as a result of the dissolved ionic solutes in it. It is measured on a scale from 0 to 50, 000 $\mu S/cm$. This gives the idea of the available salt electrolytes and ions dissolved in the water sample. Water with too high number of ions or electrolytes possesses threat to human health and body organs. Also, too low number of ions signifies deficiencies in the nutrients or mineral element in the water. The lower the electrical conductivity of the water, the purer the water. Low levels of salts are found naturally in waterways and are important for plants and animals to grow. High salt levels in freshwater causes problems for aquatic ecosystems and becomes complicated in human organs.

The TDS and EC were measured using standard procedure and their values were compared to the World Health Organization (WHO) standard. These estimate the quality of the desalinated water from the four water samples before and after the experiment. Good and most suitable drinking water for human has an EC range between 0-800 $\mu S/cm$, although 800-2500 $\mu S/cm$ can still be consumed but not so preferable (Bruvold WH and Ongerth HJ., 1969; International Organization for Standardization, 1985; Nash, L., 1993; WHO/UNEP, GEMS., 1989; World Health Organization (WHO), 1986, 2007b). United States Environmental Protection Agency (EPA) classifies TDS as a secondary contaminant. It is measured in milligrams per unit volume of water ($mg/L$) and referred to as parts per million ($ppm$). For drinking water, the maximum concentration level for TDS is 600 $mg/L$ although water with extremely low TDS concentrations possesses flat, insipid taste and other adverse effects on the gastrointestinal tracts in humans (Kozisek F., 2005; Nash, L., 1993; World Health Organization., 2011; World Health Organization (WHO), 1996, 1998, 2007a).

## 3 Materials and Methods

This research work was carried out in the Department of Mechanical Engineering, Obafemi Awolowo University, Nigeria (Latitude 7.5175° N and longitude 4.5270° E) between the month of July and September 2015. Two sets of experiments were prepared: the conventional solar still (CSS) and conventional solar still with a flat plate collector (CSS-FPC). For CSS-FPC type, a pressure valve was used to prevent water inlet into the still until the desired water temperature and the pressure was reached to sufficiently force the pressure valve opened to allow the flow of water to the still basin from the flat plate solar collector.

In this experimental work, the conventional solar still was fabricated with a square stainless-steel sheet of 1 m$^2$ and 2 mm thickness. The surface area of the solar collector that receives the heat from the sun measures 1 m$^2$. The solar still basin was coated with a black paint in order to increase the solar radiation absorptivity of the still. The black body absorbs the heat energy from the sun to raise the temperature and the vapor pressure of the water (Cowling, T. G., 1950; Manabe, S., & Wetherald, R. T., 1967). Figure 1 shows the isometric and the exploded view of the experimental setup, while Figure 2 shows the photo of the experimental setup. A single slope CSS was used

basically because it is a good recipient of higher levels of solar radiation at both low and high latitude stations compared to its doubled-sloped counterpart (Sivakumar and Sundaram, 2013). Stainless steel was used to construct the basin principally due to its higher heat retaining capacity and higher resistance to corrosion that could further contribute to the water salinity. The CSS exterior walls (the sides and the bottom) were thermally insulated using 5 cm fibreglass to prevent heat energy loss from the solar still to the surroundings. Silicon sealant was used to prevent water leakage within the system and to create an air-tight environment in the interior.

The solar still was covered with a condensing glass having 5 mm thickness. The glass selected was a tempered glass of high tensile strength capable of withstanding high solar radiation intensity, wind and rain load with very low solar reflectivity (El-samadony et al., 2016). Morad *et al.* ( 2015) showed that increasing the glass cover thickness reduces the amount of solar radiation that passes through it into the air gap then to the basin water hence reduction in SS thermal retention ability and efficiencies as the glass cover thickness increases. The glass inclination is one of the major parameters that determine the CSS performance. SS productivity was found to increase with a decrease in glass inclination (Edlin, 1973; Garg and Mann, 1976). In the present experiment, the tilt angle of the glass cover was kept at 17°52'', that is, the latitude, Ø of the research location (7.5175° N) plus 10° (Chinnery, 1971; Elsayed, 1989; Felske, 1978; Heywood, 1971; Khorasanizadeh et al., 2014; Qiu and Riffat, 2003; Stanciu and Stanciu, 2014). A float valve was used to maintain a constant water level in the basin as the water flows from either the storage tank or the flat plate collector. Water productivity has been found to be inversely proportional to the water depth (Elango et al., 2015a, 2015b; Kabeel et al., 2014a, 2012; Manokar et al., 2014; Muftah et al., 2014; Nafey et al., 2000). Also, a depth of  5 cm was found to be the optimum water depth for an improved SS performance according to Kabeel *et al.* [28,29]. In addition, the higher the distance between the glass cover and the basin's water surface, the more the energy and the time required of the vapor to travel to the inner glass surface (Tiwari, A. K., & Tiwari, G. N., 2005, 2008; Tiwari et al., 1994). Hence, the gap was reduced to 2 cm.

**3.1 Experimental Design**

Four different water samples (rainwater, freshly dug well water, river water and heavily polluted water) commonly consumed by people in rural settlements in Nigeria due to unavailability of clean drinkable water were selected for the purpose of this research. The water from these sources has been found to be dirty and unhygienic for human consumption (Onwujekwe et al., 2009; Smith et al., 2004). The villagers, even passers-bye have their bathe, urinate, defecate and even dispose-off their refuses or dirt in the river water and the heavily polluted water. Following the solar still design and set up above, the experiments were conducted for a period of thirty days between 8 a.m. and 6 p.m. while readings were taken on an hourly basis. One water sample was chosen for each day and was filled into the solar still basin to the required depth (5 cm as mentioned above). The basin was subsequently tilted to angle 17°52'' based on the geographical location of the research. The experiment set up was left outside in the sun to run between 8 a.m. and 6 p.m daily. During this period, the heat from the sum evaporates the water in the basin and later condenses on the inner surface of the glass which is later channeled and collected. The temperature of the inner surface of the glass (condensing surface), the outer surface of the glass, absorber plate (evaporating surface) of the solar still, basin water temperature and temperature of the glass of the flat plate collector were measured and recorded intermittently on hourly basis through a data logger while the experiment is ongoing. Five pieces of Copper-constantan thermocouples (Type T) with temperature readout were strategically mounted on different parts

of the experimental set up to measure temperatures at specific locations. Also, the most important meteorological parameters for efficient performance of the CSS such as solar radiation, ambient temperature and wind velocity were subsequently measured and recorded. By the law of nature, these parameters cannot be controlled/altered; however, they were measured using a weather station positioned at the research location (Figure 2). A transmitter was incorporated into the weather station. This was used to download and record necessary meteorological parameters such as the amount of rainfall, ambient temperature, relative humidity, wind velocity and luminous intensity. Dust deposition and shade coverage on the glass surface reduces the transmittance power of the solar radiation which could affect the distillate yield and the efficiency of the solar still. Hence, these were controlled by placing the experimental set up at a height of clean environment, a little above the ground cleared of anything that could constitute shade coverage and the glass surface also was cleaned intermittently using a wet towel.

The basin was washed and made ready for another water sample after each experiment. These experiments were conducted for both conventional and the single slope solar still with flat plate collector following the same steps discussed above.

## 3.2 Performance Evaluation

The TDS in the water sample was measured using a digital conductivity meter by Mettler Toledo with ±0.5 % conductivity accuracy. The digital meter was used to measure both the TDS and the EC. It consists of a mode which is usually interchanged/switched when either the TDS or the EC measurement is required. This digital meter consists of a probe. For each time, each water sample was to be tested, the probe was immersed into the water sample up to the maximum manufacturer's immersion level after the protective cap was removed while the temperature of the water sample is maintained at room temperature.. The water sample was thoroughly agitated to dislodge air bubbles and evenly distribute the particulate matter present in the water. The TDS and the EC level for the sample were taken after the reading stabilizes. After each measurement, the probe was thoroughly cleansed as prescribed in order to eliminate the interference of the previous sample particle with the current sample. The digital meter also displays the temperature of the water sample to be measured. The reading gives us the salinity estimate of the produced fresh water from the solar desalination unit.

The percentage reduction in TDS and EC was be calculated using Eq. (1)::

$$\% \ Reduction = \frac{P_b - P_a}{P_b} \ x \ 100\% \qquad (1)$$

P = Parameter under consideration (TDS or EC).

Subscript a and b represent after and before respectively.

The solar still instantaneous efficiency, $\epsilon_i$ was calculated using Eq. (2):

$$\epsilon_i = \frac{M \ x \ h_{fg}}{A \ x \ I \ x \ \Delta t} \qquad (2)$$

where,   M = mass of the desalinated water at the output

$h_{fg}$ = latent heat of vaporization of the fluid

A = Area of the flat plate collector (1 m$^2$)

$I$ = Average solar irradiation for the time under consideration

$\Delta t$ = Time under consideration (usually 1 hr).

Also, the daily production efficiency, $\epsilon_d$ of the solar still system was be calculated using Eq. (3):
$$\epsilon_d = \frac{\sum P_h \times h_{fg,da}}{(C A_a \times \sum I)\Delta t} \tag{3}$$
where,  $P_h$ = distillate productivity per hour $A_a$ = Absorber Area
C = Concentration ratio that is ${A_{ap}}/{A_a}$    $A_{ap}$ = Aperture Area
$h_{fg,da}$ = latent heat of vaporization daily average.

## 4.0 Results and Discussion

Experiments were conducted for a period of thirty days between 8 a.m. and 6 p.m. while readings were taken on an
hourly basis. The experiment started on the 1st of July 2015 and ended on the 17th of August 2015. Some randomly
selected results of the experiments are presented in Table 1.

### 4.1 Solar radiation and temperature variations in solar still

Solar radiation is the radiant energy emitted and deposited by the sun in an area every second from a nuclear fusion
reaction that creates electromagnetic energy with a temperature of about 5800 K. It is one of the most important
factors that determines the solar still productivity (Sharshir et al., 2016). Figure 3 (a–d) shows the variation of solar
radiation intensity, ambient temperature, glass temperature, absorber plate temperature and water temperature with
time for some randomly selected days. The graphs and results for other days share some similarities. It was observed
that the temperature keeps increasing until maximum point around 3 pm in the afternoon for all days of the
experiment. This is due to a consistent daily increase in the solar radiation intensity until 3 pm in the afternoon. The
temperatures begin to drop as soon as the solar radiation intensity begins to drop, and vice versa. This shows that the
solar radiation intensity determines the temperatures of the elements in the still. It was also observed that the
ambient temperature is always lower than all other temperatures for all days of the experiments in the research
location. The solar radiation was maximum on the first day of the experiment with the intensity of about 1128 W/m$^2$
at 3 pm in the afternoon and the lowest value obtained was 27.2 W/m$^2$ on the second day of the experiment at 7 am
in the morning. The solar radiation intensity was measured with Eppley precision spectral pyrometer (PSP) with an
accuracy of ±0.5% from 0 to 2800 W/m$^2$.
It was observed that the evaporation rate and consequently the distillate yield increases as a result of an increase in
the temperature difference between the temperature of the inner surface of the glass (condenser) and the temperature
of the absorber plate (evaporator). From the graphs in Figure 4, it could be depicted that the glass temperatures are
far lower than the temperature of the water. The minimum condensation glass temperature obtained was 25 $^o$C and
the maximum was 40 $^o$C. The wind speed of the environment at the moment under consideration affects the rate of
condensation by the glass. The faster the wind speed the faster the vapour loses its latent heat of vaporization to the
surroundings. The increased wind speed yields a rapid drop in the condensing glass temperature and hence a wide
temperature difference between the condensing glass and water. This enhances the heat transfer performance and
hence the distillate yields because heat transfer rate is directly proportional to temperature difference. This is in good

agreement with some similar past studies (El-Sebai, 2000; El-Sebaii, A. A., 2004; Stonebraker et al., 2010; Winfred Rufuss et al., 2017).

The temperature increase in the absorber shows that the absorber and the black body material is a good absorber and retainer of heat. This property is responsible for evaporation even in off-peak periods when there is no sunlight and little or no solar irradiance. The stored heat in the black body raises the temperature of the water in the basin and with the corresponding saturation pressure, evaporation occurs. The maximum temperature obtained for the absorber was 63 $^{o}$C

## 4.2 Effect of temperature variation on distillate yield

Figure 4(a–d) shows the effect of temperature variation on distillate yield. Figures 4 a and d gives the distillate yield for the active solar still while Figures 4b and c represent the distillate yield for the passive still. The graphs also justify that temperature difference (that is the difference between the glass cover and absorber plate temperatures) is the major factor responsible for evaporation. This trend was also observed by several authors but to mention a few (Ahsan et al., 2013; Ali et al., 2019; Edeoja et al., 2015; Kumar and Bai, 2008; Murugavel et al., 2010; Onyegegbu, 1986; Ozuomba et al., 2017; Sathyamurthy et al., 2015). Preheating the feed water to the solar still basin plays an important role in increasing the productivity of the still (Ahmadi et al., 2017; Badran and Abu-khader, 2007; Delgado-Torres et al., 2007; Kalogirou et al., 2016). Comparatively, huge distillate yield was experienced when the flat plate collector was used on days 8, 1, 9 and 2 as shown in Figures 6 (a and b). The solar still was used alone without the flat plate collector in the remaining days. It was observed that continuous deposition of hot water into the basin from the Flat Plate Collector resulted into higher production rates in all operation periods and mainly between 2–4 pm daily. This is due to higher internal convective, evaporative and radiative heat transfer from the water to the glass cover as the preheated water from the flat plate solar collector is deposited to the basin. Higher temperature differences were observed in solar still with the flat plate collector compared with that of no flat plate collector throughout the working hours and under all conditions of the experiment.

## 4.3 Effect of solar radiation on distillate yield

Figure 5 (a–d) shows the variation of solar radiation intensity and the distillate yield with time. Like the temperatures, the solar radiation intensity had a similar effect on the distillate yield. However, the differences between the effects with and without the flat plate collector cannot be easily detected using the solar radiation intensity curve alone. The temperature curves clearly show the differences between the glass temperature and the water temperatures and their consequential effects on the solar still productivity. Furthermore, the graphs (Figure 5 a–d) clearly indicate that the incident solar radiation strongly determines the increase in the Still productivity.

## 4.4 Cumulative distillate yield and the hourly distillate yield

Figure 6 a and b present the cumulative distillate yield and the distillate yield per hour for the 9 days, respectively. The graphs clearly show the significant differences between the cumulative yield and the distillate yield per hour of the still incorporated with the Flat Plate Collector and the ones without the Flat Plate Collector. Day 8 shows significant cumulative distillate yield not only because of the second largest solar radiation intensity recorded for the day (965 W/m$^2$) but basically because of the comparative huge temperature difference between the glass cover

(condensation surface) and the water in the basin and the consistently higher solar radiation intensity recorded for the other hours of the day. As earlier discussed, the variations observed in the distillate yield are due to the condensation glass-water temperature difference, wind speed variations and relative humidity of the research location per time. The contents of the polluted/saline water and the extents at which the water is polluted also affects the evaporation rates and hence the solar still productivity because the presence of impurities increases the boiling point of a fluid (or any substance) (Cengel, Yunus A., and A. J. Ghajar., 2011). Details of this are not explored in this research.

**4.5 Laboratory Examination of the Water Samples before and after Desalination        (Quality of the distillates from the raw water samples)**

Table 2 shows the results of the water analyses conducted before and after the solar distillation process. Observation shows that water quality lies within the acceptable range for good and drinkable water according to WHO prescription for EC and TDS. Also, the physical appearance of the distillate/desalinated water shows good turbidity (water looks so clear and colorless) appealing for human consumption. Also, the repulsive and the irritating odor of the heavily polluted water was drastically reduced.

**4.6 Comparison of the TDS and the EC readings obtained against existing results**

The TDS and the EC of the produced desalinated water from the four difference sources has been compared with some results available in the literature of various solar still with different configurations of solar desalination system (Table 3).

**4.7 Comparison of the distillate yield in the present studies against that which exist in the literature**

Several authors have worked on performance evaluation of solar still of different configurations. Their results are hereby compared with that of the present studies. With the understanding that the performance of any solar still is dependent on the location under consideration viz-a-vis the inherent/current climatic and atmospheric condition, diurnal irradiance and other specified experimental conditions, however, it can be noticed that the performance of the solar still in consideration is relatively comparable with those existing in the literature and in some cases of better performance despite the simple design.

**4.8 Solar Still Efficiency**

The average of the overall daily efficiencies of the conventional solar still with flat plate collector and the single slope solar still with flat plate collector are 13.906 % and 16.298 % respectively. This shows an improvement of 14.67 % with the inclusion of the single slope design compared with the conventional type. Since these values are dependent on the weather, climate and the atmospheric conditions with the diurnal irradiance coupled with the still design, hence it is difficult to compare with existing designs in the literature.

The daily production efficiency, $\epsilon_d$ of the still are 15.85 % and 26.25 % respectively for the conventional solar still with flat plate collector and the single slope solar still with flat plate collector.

**4.9 Cost**

It is important to estimate the cost of solar still basically for the purpose of improvement both in terms of production and efficiency. Kabeel et al. (2010) listed the running and capital costs that affects the cost of production of a solar still such as design and size of the unit, climatic condition of the site, the properties of the feed water, the required quality of the distilled water to be produced and the cost of wages for available staff.

The adopted design in this research is tailored towards cost effective and simple infrastructure produced from locally sourced material which are readily available, easily produced, operate and maintained. This is ensured so that the set up can easily be acquired by an average family in the rural areas to make portable water readily accessible.

The solar still in this present study is made with locally sourced materials and as at the time of the construction the average cost is approximately $ 150. The analysis for the cost per liter of distilled water based on Kabeel et al. [62] is as follows:

Passive solar still

Fixed Annual Cost FAC = 40 USD

Annual Salvage Value ASV = 7 USD

Annual Maintenance Cost = 2 USD

Annual Cost AC = FAC + AMC – ASV = 35 USD

Annual Productivity M = 1.154 kg/ $m^2$ = 421 litres/year $m^2$

Active solar still

Fixed Annual Cost FAC = 140 USD          (100USD cost of flat plate collector)

Annual Salvage Value ASV = 70 USD

Annual Maintenance Cost AMC = 5 USD

Annual Cost AC = FAC + AMC – ASV = 75 USD

Annual Productivity M = 2.396 kg/ $m^2$ = 874.54 litres/year $m^2$

Cost of Distil Water per litre CPL = AC/M

CPL (Active) = 0.0858 USD/ltr

CPL (Passive) = 0.0831 USD/ ltr

Compare the cost per litre of distilled water by the present design with earlier designs by Kumar and Tiwari [101], Badran and Tahaineh [102], Abdallah and Badran [91],  the present design showed a significant reduction in cost of production and can be adopted by rural communities that are have shortage of drinkable water.

**5 Conclusion**

The possibility of using the renewable energy from the sun in providing potable drinkable water from saline or heavily polluted water in areas where potable water is scarce has been explored using solar desalination technology. Solar desalination method has been found to be a clean energy and eco-friendly, readily accessible, affordable, easy and renewable method of purifying water. A single slope rectangular basin was designed and constructed with low cost, lightweight, available locally sourced materials. The effects of solar radiation intensity, ambient temperature, condensing inner glass cover temperature, water temperature and absorber temperature on the water distillate yield from the solar still were observed based on the climatic condition of Ile-Ife, Nigeria. Results show the direct

relationship and huge dependency of solar still daily distillate yield on the solar radiation intensity and the temperature difference between the condensing inner glass cover and the water. A high distillate yield was recorded when the solar radiation intensity was at the peak accompanied with temperatures increase for all the solar still components at the same time in the day. The temperatures increased as the solar radiation intensity increased, however, the larger increase was experienced for water and the absorber in the basin, this was primarily due to the heat retaining ability property of the black body used. The wind speed of the research station also was a contributing factor to the drop in the glass temperature, hence constituting a huge temperature difference between the condensing inner glass cover and the water for higher heat transfer and evaporation rate and larger distillate yield. The impact of the flat plate collector on the distillate yield was also investigated. The incorporation of the flat plate collector produced higher distillate yield. The preheated water it supplied created a huge temperature difference between the condensing inner glass cover and the water which consequentially produced more distillate yield compared to a solar still without flat plate collector. The desalination product quality was analyzed based on its electrical conductivity and the amount of total dissolved solid present in it. The distilled water was found to be within the acceptable range for drinkable water according to the World Health Organization standard and guidelines. This shows the potential of water desalination using solar energy most especially in areas where water-borne diseases are imminent due to the scarcity of potable drinkable water. It could be predicted from the results trends that the distillate yield would be higher during the dry season characterized by higher solar radiation intensity compared to the solar radiation intensity recorded during the raining season during the period in which the experiment was performed.

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

**TABLES**
**Table 1: Experimental Set-up for the Desalination**

| S/N | Date | Type of Sample | Type of Solar Still |
|-----|------|----------------|---------------------|
| 1 | 02/07/2015 | River Water | Active |
| 2 | 06/07/2015 | Rainwater | Active |
| 3 | 10/07/2015 | Dug-well Water | Passive |
| 4 | 14/07/2015 | Heavily Polluted Water | Passive |
| 5 | 15/07/2015 | Rainwater | Passive |
| 6 | 25/07/2015 | Heavily Polluted Water | Passive |
| 7 | 27/07/2015 | Dug-well Water | Passive |
| 8 | 05/08/2015 | Heavily Polluted Water | Active |
| 9 | 10/08/2015 | Dug-well Water | Active |


**Table 2: Water Analyses results before and after desalination**

| Water Sample | TDS (mg/liter) or (ppm) | | Electrical Conductivity ($\mu$S/cm) | |
|--------------|------------------------|---|-------------------------------------|---|
| | Before distillation | After distillation | Before distillation | After distillation |
| Rainwater | 19 | 14 | 14 | 23 |
| Freshly dug well water | 97 | 21 | 162 | 35 |
| River water | 75 | 36 | 125 | 60 |
| Heavily polluted dirty water | 143 | 13 | 238 | 22 |
| WHO Standard | < 600 mg/L | | 0-800 $\mu S/cm$ | |


**Table 3: Performance comparison of the solar still in terms of TDS and EC reduction**

| S/N | Authors | Type of Solar Still | Type of water | % Reduction in TDS | % Reduction in EC |
|---|---|---|---|---|---|
| 1 | Present study | Flat plat collector | Rainwater | 26.316 | 64.29 |
| | | | Freshly dug well water | 78.351 | 78.395 |
| | | | River water | 52 | 52 |
| | | | Heavily polluted dirty water | 90.909 | 90.756 |
| 2 | Samee et al. [81] | Single basin solar still | Simly dam filtration plant water | 91.89 | 96.82 |
| 3 | Kumar and Bai [72] | Basin type solar still with improved condensation technique | Tap water | 74.23 | 81.87 |
| | | | Seawater | 99.61 | -256.58 |
| | | | Dairy effluent | 84.95 | -5160.00 |
| 4 | Flendrig et al. [82] | Thermoformed solar still | Contaminated water source | 98.48 | 99.64 |
| 5 | Arunkumar et al. [83] | Hemispherical solar still | Water | 87.50 | 90.00 |
| 6 | Omara et al. [84] | Hybrid desalination system using wicks/solar still and evacuated solar water heater | Water | 89.21 | - |
| 7 | Ahsan et al. [74] | Triangular solar still | Seawater water | 73.25 | 73.25 |
| 8 | Nagarajan et al. [85] | Triangular Pyramid Solar | Fresh Water | 89.58 | 92.57 |
| | | | Synthetic water | 87.04 | 87.04 |
| | | | lab-prepared water | 98.72 | -3.75 |

 

| S/N | Authors | Type of Solar Still | | Maximum daily productivity ($day/m^2$) |
|-----|---------|---------------------|---|---------------------|
| 1 | Present Study | Active still | | 2.396 kg |
| | | Passive still | | 1.154 kg |
| 2 | Voropoulos et al. [86] | Still coupled with solar collectors | | 4.2 kg |
| 3 | Boukar and Harmim [87] | One-sided vertical solar still | | 1.4 kg |
| 4 | Tiwari et al. [88] | Flat Plate Collector | | 0.500 kg |
| 5 | Tarawneh [89] | Conventional Still | | 0.720 kg |
| 6 | Badran and Abu-khader [78] | Single slope solar still | 3.5 cm depth | 0.590 kg |
| | | | 2.0 cm depth | 0.800 kg |
| 7 | Velmurugan et al. [90] | Solar still with fin | | 0.425 kg |
| 8 | Abdallah and Badran [91] | Fixed and Tracking solar stills | | 0.175 kg |
| 9 | Singh et al. [92] | Hybrid photovoltaic thermal (PVT) double slope active solar still | Series | 1.07 kg |
| | | | Parallel | 1.30 kg |
| | | | Natural | 0.90 kg |
| 10 | Omara et al. [84] | Conventional | | 0.44 kg |
| | | Single layer lined wick | | 1.00 kg |
| | | Single Layer square wick | | 1.10 kg |
| | | Double layer lined wick | | 0.78 kg |
| | | Concentrating Collector | | 0.6 kg |
| | | Evacuated Tube Collector | | 0.64 kg |
| | | Evacuated Tube Collector with heat pipe | | 0.70 kg |
| 11 | Ahsan et al. [74] | Triangular Solar still | 1.5 cm depth | 0.04 kg |
| | | | 2.5 cm depth | 0.05 kg |
| | | | 5.0 cm depth | 0.033 kg |
| 12 | Gorjian et al. [93] | Stand-alone point-focus parabolic solar still | | 1.07 kg |
| 13 | Omara et al. [94] | Stepped solar still | | 1.18 kg |
| | | Conventional | | 0.65 kg |
| 14 | Elango and Murugavel [95] | Double basin stills | | 0.525 kg |
| 15 | Sathyamurthy et | Still without PCM | | 0.22 kg |

| | | | | |
|---|---|---|---|---|
| | al. [73] | Still with PCM | | 0.12 kg |
| 16 | El-Agouz et al. [96] | Continuous flow inclined solar still | | 0.6 kg |
| 17 | Elango et al. [95] | Single slope solar still with different water nanofluids | Water | 0.092 kg |
| | | | Water + $Al_2O_3$ | 0.160 kg |
| | | | Water + ZnO | 0.125 kg |
| | | | Water + $SnO_2$ | 0.132 kg |
| 18 | Kumar and Rajesh [97] | Hybrid still | | 0.62 kg |
| 19 | Faegh and Behshad [98] | Solar still with PCM | | 1.03 kg |
| 20 | Panchal and Mohan [99] | Conventional solar still | | 0.390 kg |
| | | Circular fin solar still | | 0.520 kg |
| | | Square fin solar still | | 0.590 kg |



**FIGURES**

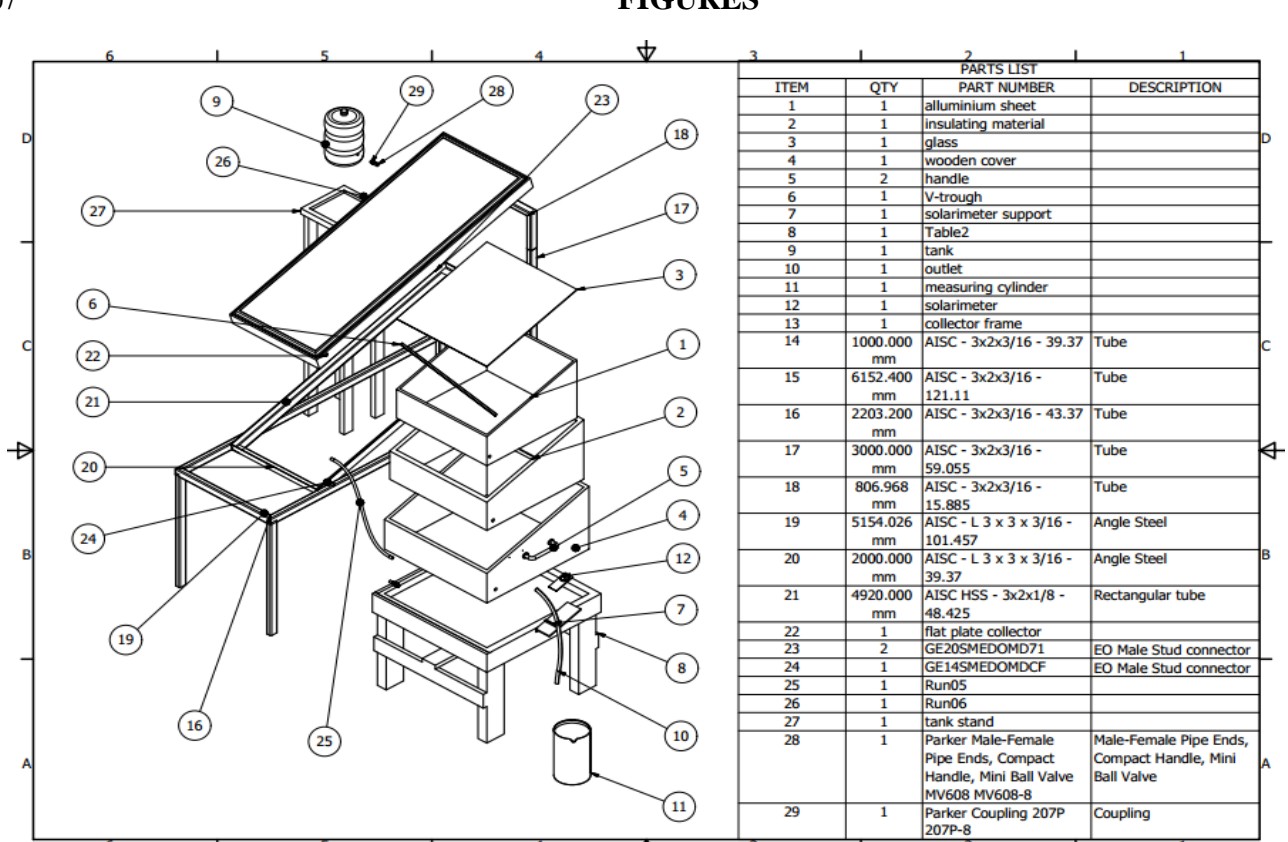

| PARTS LIST | | | |
|---|---|---|---|
| ITEM | QTY | PART NUMBER | DESCRIPTION |
| 1 | 1 | alluminium sheet | |
| 2 | 1 | insulating material | |
| 3 | 1 | glass | |
| 4 | 1 | wooden cover | |
| 5 | 2 | handle | |
| 6 | 1 | V-trough | |
| 7 | 1 | solarimeter support | |
| 8 | 1 | Table2 | |
| 9 | 1 | tank | |
| 10 | 1 | outlet | |
| 11 | 1 | measuring cylinder | |
| 12 | 1 | solarimeter | |
| 13 | 1 | collector frame | |
| 14 | 1000.000 mm | AISC - 3x2x3/16 - 39.37 | Tube |
| 15 | 6152.400 mm | AISC - 3x2x3/16 - 121.11 | Tube |
| 16 | 2203.200 mm | AISC - 3x2x3/16 - 43.37 | Tube |
| 17 | 3000.000 mm | AISC - 3x2x3/16 - 59.055 | Tube |
| 18 | 806.968 mm | AISC - 3x2x3/16 - 15.885 | Tube |
| 19 | 5154.026 mm | AISC - L 3 x 3 x 3/16 - 101.457 | Angle Steel |
| 20 | 2000.000 mm | AISC - L 3 x 3 x 3/16 - 39.37 | Angle Steel |
| 21 | 4920.000 mm | AISC HSS - 3x2x1/8 - 48.425 | Rectangular tube |
| 22 | 1 | flat plate collector | |
| 23 | 2 | GE20SMEDOMD71 | EO Male Stud connector |
| 24 | 1 | GE14SMEDOMDCF | EO Male Stud connector |
| 25 | 1 | Run05 | |
| 26 | 1 | Run06 | |
| 27 | 1 | tank stand | |
| 28 | 1 | Parker Male-Female Pipe Ends, Compact Handle, Mini Ball Valve MV608 MV608-8 | Male-Female Pipe Ends, Compact Handle, Mini Ball Valve |
| 29 | 1 | Parker Coupling 207P 207P-8 | Coupling |

**Figure 1: Isometric diagram and the exploded view of the experimental setup.**


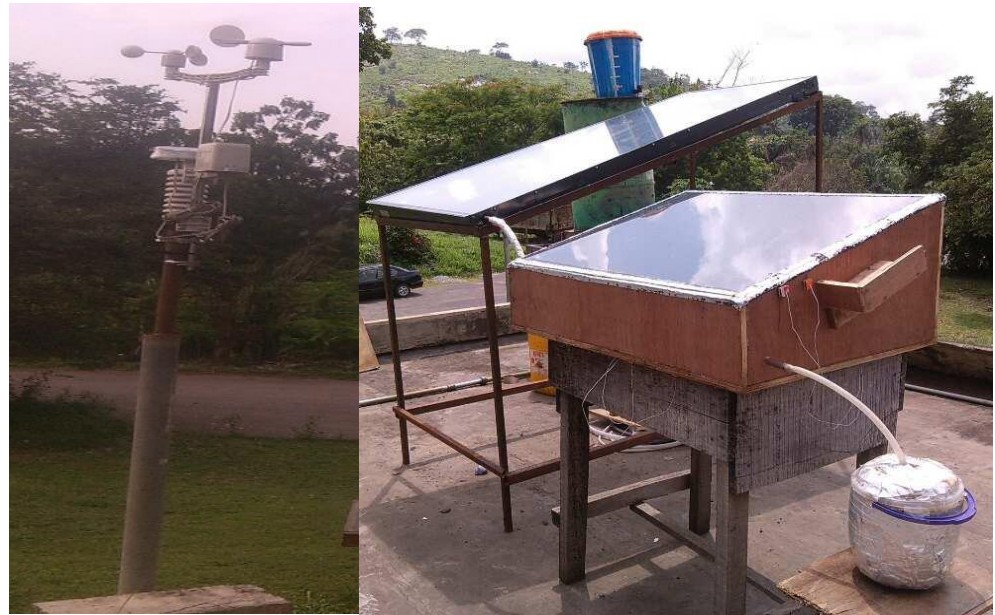

**Figure 2: Experimental set up of solar still coupled with flat plate collector**


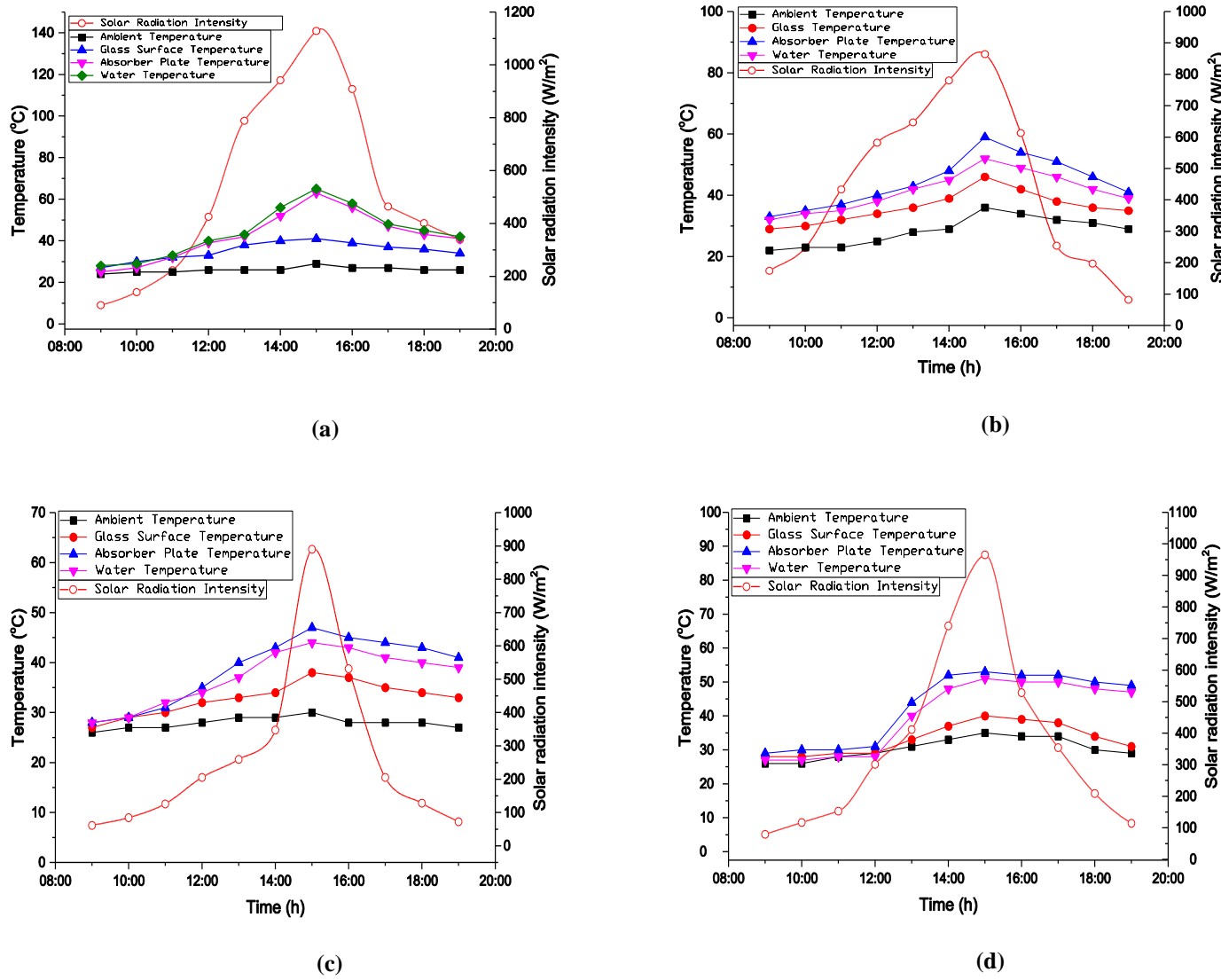

**Figure 3: Daily temperature variation with solar radiation intensity (a) day 1 (b) day 3 (c) day 6 and (d) day 8**

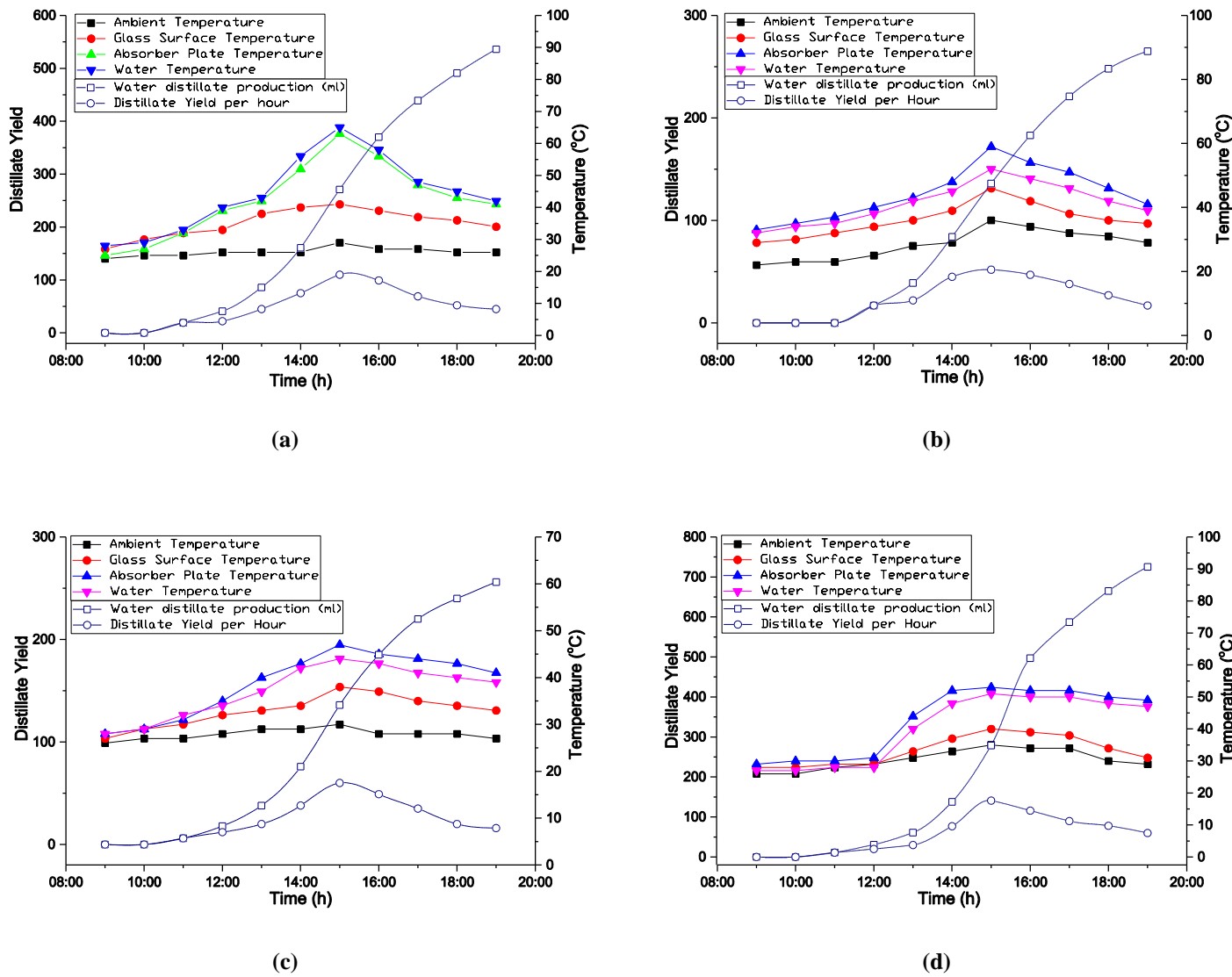

Figure 4: Influence of temperature on distillate yield (a) day 1 (b) day 3 (c) day 6 and (d) day 8



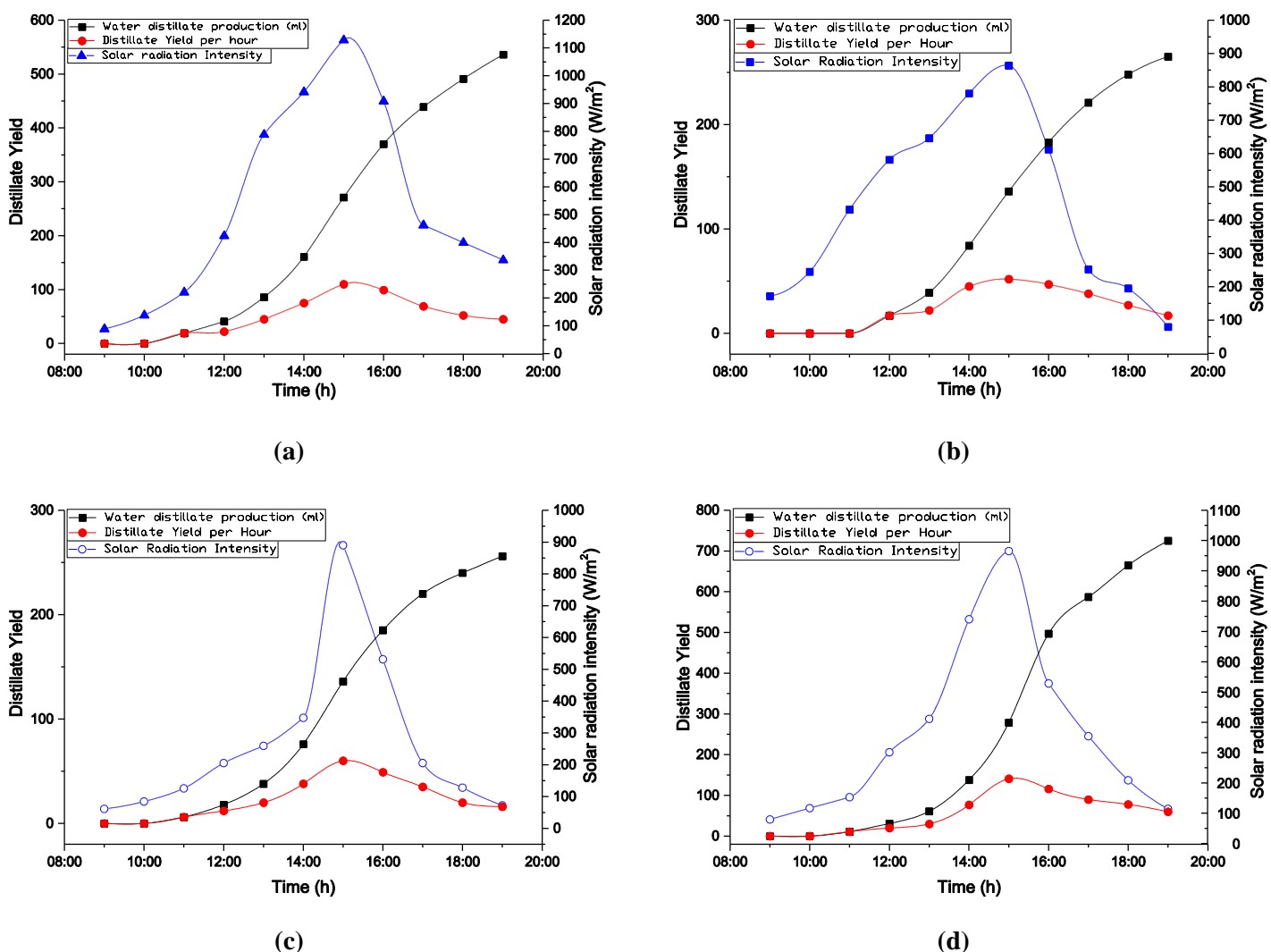

**Figure 5: Influence of solar radiation intensity on the distillate yield (a) day 1 (b) day 3 (c) day 6 and (d) day 8.**

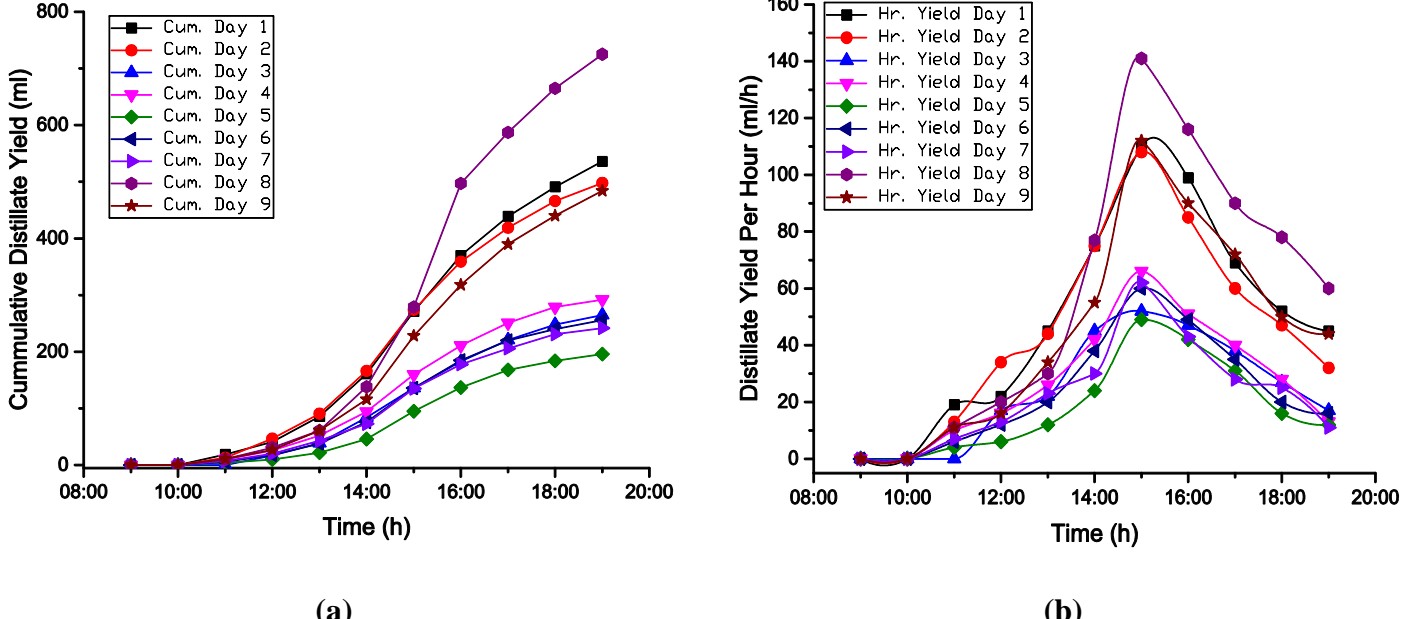

(a)             (b)

**Figure 6: Distillate yield (a) Cumulative distillate yield (b) Distillate yield per hour**

**AUTHOR'CONTRIBUTION**

**Saheed A. Adio**
Conceived the idea, defined the problem statement, the main supervisor on the project and did several rounds of reviews during the writing stage of the manuscript.

**Emmanuel A. Osowade**
Experimental setup, data collection and wrote the introduction and literature review and the cost analysis section

**Adam O. Muritala**
One of the project co-supervisors, and worked on the problem definition and several rounds of reviews during the writing stage of the manuscript

**Adebayo A. Fadairo**
Experimental setup and data collection. Also, worked on the data analysis and graphical representations.

**Kamar T. Oladepo**
One of the project co-supervisors. He worked on the experimental design and results interpretations.

**Surajudeen O. Obayopo**
Project supervision during the experimental setup and data collection, and the review of the manuscript after first completion.

**P. Fase**
Experimental setup and data collection and some initial write-ups.

## COMPETING INTERESTS

This is to confirm that there are no known conflicts of interest associated with this publication
and there has been no significant financial support for this work that could have influenced its
outcome. We confirm that the manuscript has been read and approved by all named authors and
that there are no other persons who satisfied the criteria for authorship but are not listed.