# Peer review of "Solar Distillation of Impure Water from Four Different Water Sources under South-Western Nigeria Climate"

_Drinking Water Engineering and Science, 2020_

## Referee Comment (RC1) · Bas Heijman (Referee) · 26 Jun 2020

In the paper the results are presented of experiments with a home build solar still. The experiments are monitored well and the results are useful. But there are some parts of the paper that need clarification.

s10: rephrase this line (havoc is not the appropriate terminology). For instance: The major problems caused by scarcity of drinking water

s20: TDS-measurment is in this case probably measured with electrical conductivity. The meter is measuring the EC and a linear relation is assumed between EC and TDS.

[Figure]

So on the display there a a TDS-reading. This linear relation is not very accurate. The proper way to measure TDS is to measure all the ions in the water matrix in mg/l and add them up to obtain the total amount of dissolved ions. So I would suggest only to talk about EC in this paper.

s51 here electrolysis is mentioned but this is a process to split water in H2 and O2. What is meant is electrodialysis. This is a process with two different membranes and two electrodes pulling ions trough the membranes resulting in one diluate (less ions) and one concentrate (more ions) stream.

s90: Why is roof water carcinogenic? I can imagine that it contains bacteria and viruses (from the birds on the roof) and it can contain metals like iron and zinc from the roof material but I cannot imagine that it contains carcinogenic compounds. If this is possible please refer to literature to prove this.

s96 refrase "as a result of indiscriminate drinking of water". Drinking water according to the WHO-guidelines will never contain water born diseases. So probably you mean that people use water that is not treated to drinking water or the quality is not meeting the WHO-guidlines.

S106 mention here also the m2 of the solar collector because this surface area contributes to the solar heat that is collected during the experiment.

S207 avoid terminology like "ridiculous"

S210 mention the brand of the measuring equipment (but the TDSmeter is in fact a conductivity meter)

s281 mention which graph is the active and which graph is the passive setup (probably: a and d are active and b and c are passive setups)

s323: mention that EC and TDS-removal rate is not very relevant in this case because the starting TDS is already below the WHO-guidlines. If seawater or brackish water was investigated this was a more relevant parameter. And for seawater the reduction

rate should be something like 99.9% to obtain drinking water.

s336 the table here (tabel has no number!!!) shows a unit I cannot understand: Maximum daily production rate (kg/m2hr). So probably the proper unit is kg/(m2.day) In the table you should mention for comparison your results for the passive setup and the active setup. And mention if the m2 of the solar collector is used in this calculation. Becasue in fact you should refer the production to the total m2 surface area you use to collect solar heat.

s346 The XXXX should be replaced by numbers??? Please do so. Otherwise delete this part.

---

## Author Comment (AC1) · 1 Jul 2020

S10: We thank the reviewer for the good suggestions. The observation has been rectified as suggested. s20: We use a digital meter that measures both the TDS and the EC. The modes were only interchanged to capture the required parameter at each specific time. As reported in lines 217-220, The total dissolved solid particles in the water sample was measured using a digital conductivity meter by Mettler Toledo with $\pm 0.5$ % conductivity accuracy. However, emphasis is reduced on TDS due to the suggestion made by the reviewer (RC1). S51: We appreciate the reviewer for this observation. The suggestions has been effected S90: This observation has been

reported in the literature as cited in line 92. S96: The sentences has been rephrased as suggested by the reviewer (Line 99 - 100) S106: The reviewer pointed out the need to mention the area of the solar collector that received the heat from the sun. This area is 1 square meter as mentioned in line 110 - 111. S207: We thank the reviewer for this observation and it has been corrected. S210: The brand of the measuring equipment is a digital conductivity meter by Mettler Toledo with $\pm0.5$ % conductivity accuracy. The digital meter was used to measure both the TDS and the EC. This is described in lines 217 - 222. S281: Figures 4 a and d gives the distillate yield for the active solar still while Figures 4b and c represent the distillate yield for the passive still. This is captured in Lines 280 - 282 s323: We appreciate the suggestions of the reviewer and this line has been modified. s336 The tables here have been numbered and the data are presented in SI units. Daily production rate is now written in litres per square meter of solar collector per day. This explanation addressed the observations raised by the reviewer. s346 The XXXX has been replaced by numbers as suggested by the reviewer. Other relevant information has been added to improve the quality of the paper.

---

## Referee Comment (RC2) · Anonymous Referee #2 · 22 Jul 2020

The removal of TDS and EC was studied using a locally made solar distillation installation. Various water sources were used and compared to other experiments described in literature. The paper is reasonably well written, but lacks a clear objective and a good discussion of the results with literature. General comments: - A clear objective at the end of the introduction is missing. How does this relate to previous research in the area? What is novel? Only location is not sufficient.. Is the design novel? - It should be explained why solar stills are used to treat the water mentioned water sources. Probably there are more cost effective ways to treat groundwater, rain water and surface water. - EC and TDS is not sufficient to judge the treatment performance since these are not indicators for microbial contamination e.g. - Comparing EC and

TDS to WHO guidelines is not sufficient to judge performance. - The discussion with literature should be included in the sections describing the results (now they are separated). - Cost analyses should be made in comparison with the production, so xxx €m3 - Language, including tenses, should be checked - Redundant information should be deleted. - Avoid too general introduction. Specific comments: - Line 21-22 and 23-26: see comment above - Line 28-38: see general comment above and delete. - Line 39: of = in; sporadic? - Line 46: which part of the world? - Line 48-51: see general comment above and delete - Line 54-57: see general comment above and delete - Line 58: bold statement; term = terms. - Line 60-62: see general comment above and delete - Line 80-81: how is this reflected in this study? - Line 84: faced = facing - Line 81 and onwards: give more references - Line 89 and onwards: explain how this "solution" relates to more economic and sustainable solutions? Or phrase the study differently.. - Line 95-96: bold statement and should be rephrased - Line 103: flow = flows - Line 118: is = was - Line 125: indicated by whom? Give references - Line 138-145: should be rephrased based on general comments, since there are many flaws in this reasoning. - Line 162: can be = was - Line 167: is = was - Line 168-173: redundant information - Line 174: give overview of all experimental settings and explain if duplicates in experiments and water sampling were performed. - Line 175: "Performance evaluation" should be part "Materials and Methods" - Line 183-186: explain what design variables were varied and evaluated for optimized performance - Line 196-209: should be rephrased (or deleted) based on the general comments above - Line 210: dissolved solids are not "particles"; What is a "digital TDS meter", (type/measurement method, etc.)? - Line 219-220: redundant information - Line 222:226: should be more extensive and part of Materials and Methods section (see comment line 174). - Line 228-230: consider deleting - Line 232: why randomly selected days? Is there another way to present all days? - Line 234/235: till = until - Line 236-237: is this relationship known from literature, then discuss this with literature... - Line 238-240: rephrase or delete - Line 240: smaller = lower - Line 243: evening = afternoon - Line 243 and onwards: how does it compare with other studies? Give references... - Line 248: than the temperature of the water - Line 256 and onwards: how does it compare with other studies? Give references... - Line 267-269: not new - Line 274: results = resulted - Line 275: do not use "significantly" when statistical analyses are not performed - Line 284: has = had - Line 294-307: can be deleted, because the graphs represent the same data of previous graphs and do not give extra information. - Line 311-319: poor performance analyses and lack of discussion with other studies (see also general comments above) - Line 320-337: should be incorporated in previous sections - Line 338-359: should be incorporated in previous sections and not ready yet (methods should be incorporated in Materials and Methods section) - Line 361-363: see general comments - Line 365-365: too general information - Line 387-394: too bold conclusion

―――――――――――――――――――――――

---

## Author Comment (AC2) · 16 Aug 2020

General comment The reviewer is so emphatical about we remove report on TDS and EC measurement. He stated this since we did not check the microbial level or activities. Many papers discuss TDS and EC so I guess we can leave it as it is. 1. Question EC and TDS are not sufficient to judge the treatment performance since these are not indicators for microbial contamination e.g. - Comparing EC and TDS to WHO guidelines is not sufficient to judge performance. Observation The scope of the study does not consider the level of microbial contamination in the water sample before and after the desalination. TDS and EC tested before and after desalination are just in addition

to the effect of solar insolation and temperature variations on the yield of the distillate from the constructed solar still. These were carried out to judge the performance of the constructed solar still. Other yardstick/parameter exist but not within the scope of this study. The main objective is to evaluate the performance of the Solar still based on the obtained yield, WHO standard on the TDS and EC of the output, Cost reduction (based on the locally sourced materials used in construction), etc. TDS and EC measurement are one of the ways by which Solar still performance is checked in the literature. Future work may include checking the level of microbial contamination before and after desalination. 2. Comment - Line 21-26: see comment above Correction Authors made the correction based on reviewer's comment 3. Comment - Line 28-38: see general comment above and delete. Correction Authors remove "microbiologically" in line 36. Other statements in line 28 – 38 expresses the view of the authors. 4. Comment - Line 48-51: see general comment above and delete Correction Authors assumes this as part of background information to support the study. 5. Comment - Line 54-57: see general comment above and delete Correction Amendment has been made. 6. Comment - Line 58: bold statement Correction Amended with reference 7. Comment - Line 60-62: see general comment above and delete -Line 81, 84 & 89: word replacement and explanation on how this work solve the water purification problem Correction Modified to reflect author's opinion 8. Comment – Line 95-96: bold statement and should be rephrased. "this is with a view" Correction Amended. New statement is not too bold or too assertive 9. Comment – Line 138 -145: Should be rephrased based on general comment. Correction Amended as suggested by the reviewer, relevant references are included 10. Comment – Line 162&167: bold statement and should be rephrased. "this is with a view" Correction Amended with new statement not too bold or too assertive 11. Comment – Line 174: give overview of all the experimental settings Observation Figures 1 and 2 have shown the experimental set up, the detail overview is not considered necessary in the author's opinion. Other issues raised regarding duplicates in experiments and water sampling have been captured under experimental design. 12. Comment – Line 175: Performance evaluation should be under material and methods

Correction Amended as suggested by the reviewer

13. Comment - Line 183-186: explain what design variables were varied and evaluated for optimized performance

Reply In this research, none of the properties mentioned in the session were varied or evaluated for optimization. All will do was that we compared the performance of passive flat plate collector against the active type. 14. Comment - Line 196-209: should be rephrased (or deleted) based on the general comments above Reply I have checked this; I see no reason to rephrase or delete. It is important to the article in my own opinion. 15. Comment - Line 222-226: should be more extensive and part of Materials and Methods section Reply This has been elaborated in the material and method section according to comment line 174. The one here is just a preamble (just a paraphrase) to the new section. 16. Comment Line 228-230: consider deleting Reply I see no reason to delete this 17. Comment - Line 232: why randomly selected days? Is there another way to present all days? Reply Experiments were carried out on several days. But we can present all the results because of space. And the results equally behave the same in as much as the solar radiation for the days under consideration look similar and it is the same experimental condition and water sample. In some case some experiments were even repeated. So, the 9 days selected are 4 days for active solar still and 5 days for the passive type. 18. Comment -Line 236-237: is this relationship known from literature, then discuss this with literature -Line 238-240: rephrase or delete Reply Deleted; not necessarily 19. Comment - Line 243 and onwards: how does it compare with other studies? - Line 256 and onwards: how does it compare with other studies? Reply Reference given 20. Comment - Line 294-307: can be deleted, because the graphs represent the same data of previous Reply Authors feel we can retain this

Please also note the supplement to this comment:
https://dwes.copernicus.org/preprints/dwes-2020-5/dwes-2020-5-AC2-supplement.pdf

---

## Author Comment (AC3) · 13 Sep 2020

S10: Rephrase this line (havoc is not the appropriate terminology). For instance: The major problems caused by scarcity of drinking water

Response: Rephrased as highlighted in line 10-12

S20: TDS-measurement is in this case probably measured with electrical conductivity. The meter is measuring the EC and a linear relation is assumed between EC and TDS. So on the display there is a TDS-reading. This linear relation is not very accurate. The proper way to measure TDS is to measure all the ions in the water matrix in mg/l and

add them up to obtain the total amount of dissolved ions. So I would suggest only to talk about EC in this paper.

Response: A digital meter that measures both the TDS and the EC was used. The modes were only interchanged to capture the required parameter at each specific time. As reported in lines 224-227. The total dissolved solid in the water samples was measured using a digital conductivity meter by Mettler Toledo with $\pm 0.5$ % conductivity accuracy. However, emphasis has been reduced on TDS due to the suggestion made by the reviewer (RC1).

S51: Here electrolysis is mentioned but this is a process to split water in H2 and O2. What is meant is electrodialysis. This is a process with two different membranes and two electrodes pulling ions trough the membranes resulting in one diluate (less ions) and one concentrate (more ions) stream.

Response: The suggestions have been effected as suggested by the reviewer and highlighted in lines 51-55.

S90: Why is roof water carcinogenic? I can imagine that it contains bacteria and viruses (from the birds on the roof) and it can contain metals like iron and zinc from the roof material but I cannot imagine that it contains carcinogenic compounds. If this is possible please refer to literature to prove this.

Response: This observation has been reported in the literature as cited in line 75-77.

S96: Rephrase "as a result of indiscriminate drinking of water". Drinking water according to the WHO-guidelines will never contain water borne diseases. So probably you mean that people use water that is not treated to drinking water or the quality is not meeting the WHO-guidelines.

Response: This has been effected as highlighted in lines 94-96.

S106: Mention here also the m2 of the solar collector because this surface area contributes to the solar heat that is collected during the experiment.

Response: The reviewer pointed out the need to mention the area of the solar collector that received the heat from the sun. This area is 1 square meter as described and highlighted in lines 158 - 161.

S207: Avoid terminology like "ridiculous"

Response: It has been corrected

S210: Mention the brand of the measuring equipment (but the TDS meter is in fact a conductivity meter)

Response: The brand of the measuring equipment is a digital conductivity meter by Mettler Toledo with $\pm 0.5$ % conductivity accuracy. The digital meter was used to measure both the TDS and the EC. This is described in lines 224 - 227.

S281: Mention which graph is the active and which graph is the passive setup (probably: a and d are active and b and c are passive setups)

Response: Figures 4 (a and d) gives the distillate yield for the active solar still while Figures 4(b and c) represent the distillate yield for the passive still. This is captured as highlighted in lines 298-300.

S323: Mention that EC and TDS-removal rate is not very relevant in this case because the starting TDS is already below the WHO-guidelines. If seawater or brackish water was investigated this was a more relevant parameter. And for seawater the reduction rate should be something like 99.9% to obtain drinking water.

Response: We appreciate the suggestions of the reviewer and this line has been modified as highlighted in lines 341-346.

S336: The table here (label has no number!!!) shows a unit I cannot understand: Maximum daily production rate (kg/m2hr). So probably the proper unit is kg/(m2.day) In the table you should mention for comparison your results for the passive setup and the active setup. And mention if the m2 of the solar collector is used in this calculation.

Because in fact you should refer the production to the total m2 surface area you use to collect solar heat.

Response: The tables here have been numbered and the data are presented in SI units. Daily production rate is now written in litres per square meter of solar collector per day. This explanation addressed the observations raised by the reviewer.

Response: S346 The XXXX should be replaced by numbers??? Please do so. Otherwise delete this part.

Please also note the supplement to this comment:
https://dwes.copernicus.org/preprints/dwes-2020-5/dwes-2020-5-AC3-supplement.pdf

**Supplement:**

| TOPICAL EDITOR'S COMMENTS | AUTHOR'S RESPONSE: Relevant lines are Highlighted in yellow colour in the main article |
| --- | --- |
| **Why did you choose to distill water with already low salt contents? (Other treatment systems are more efficient then), better to focus on seawater.** | Apart from the coastal region of Nigeria where people are forced by circumstance to process salty water for domestic use, the commonly available water in some rural areas is not pure due to dissolved organic and inorganic materials. In some locations (e.g Ile-ife Osun state: 7.4905°N, 4.5521°E) ), the economic power of the indigenes is very low and living standard is very poor. Therefore, the present work aimed at providing indigenous distillation method for locals in order to make drinkable water available at low cost. Justification is highlighted in lines 78-84. |
| **What is new in relation to the already existing solar distillators?** | Although solar distillation is not a new technology, likewise the method/structure of solar still (that is single slope conventional type) adopted in this research. However, the experimental design and the setup are location specific. The tilt angle of the glass condenser which significantly affects the output of the solar still are chosen based on the latitude of the research location, in this case 7.5175° N. Hence, the glass cover was kept at 17°52'', (i.e. the (7.5175° N) plus 10°)…….Highlighted in lines 106-109 |
| **Explain in materials and methods section your experimental design, what do you mean with "dirty water", and how the analyses were done (instruments)?** | **"Dirty water"** in this context is referred to as the water collected from stagnant water most often where passer bye (people) urinates, defecate and deposit refuse. It is heavily polluted with algae, spirogyra and refuse/dirt of all kinds. |

| | |
|---|---|
| | **"Experimental Design"** |
| | Two sets of experiments were prepared: the conventional solar still (CSS) and conventional solar still with a flat plate collector (CSS-FPC). In this experimental work, the conventional solar still was fabricated with a square stainless-steel sheet of 1 $m^2$ and 2 mm thickness. Figure 1 shows the isometric and the exploded view of the experimental setup, while Figure 2 shows the photograph of the experimental setup. The detail experimental design has been carefully explained as highlighted in lines 189-222. Analysis were carried out by first obtaining data using relevant measuring instruments and the results were analyzed in form of evaluation and comparison of results with earlier works in the literature. Important evaluation of the present work is highlighted in lines 224-252. |
| **Discuss your results in relation to literature (is it in line, is it better and why..)** | Several authors have worked on performance evaluation of solar still of different configurations. Their results are hereby compared with that of the present studies as explained and highlighted in lines 276-287, and lines 298-304. This is with the understanding that the performance of any solar still is dependent on the following factors: location under consideration (inherent/current climatic and atmospheric condition); diurnal irradiance and other specified experimental conditions.

Direct comparison of the performance of the solar still of present work with the available results in the literature is shown in Tables 3 and 4. Comparing TDS and EC, the present work is close to the |

| | earlier works in terms of TDS and EC reduction for heavily polluted water and freshly dug water which is a major problem facing the rural communities around the location where the experiments were conducted.

Other comparisons are highlighted in lines 356- 361. |
|---|---|
| **Discuss with literature the "efficiency" of the system (=efficacy and costs) in relation to other systems.** | Different methods exist for water purification using solar desalination system. The TDS and the electrical conductivity of the produced desalinated water from the four different sources have been compared with some results available in the literature using different configurations of solar desalination system. The maximum daily yield of the present work is better than most of the existing solar still as shown in Table 4.

The average of the overall daily efficiencies of the conventional solar still without flat plate collector and the single slope solar still with flat plate collector are 13.906 % and 16.298 % respectively. This shows an improvement of 14.67 % with the inclusion of the flat plate with the conventional type. Since these values are dependent on the weather, climate and the atmospheric conditions with the diurnal irradiance coupled with the still design, hence it is difficult to compare with existing designs in the literature.

The daily production efficiency, $\epsilon_d$ of the still are 15.85 % and 26.25 % respectively for the conventional solar still without flat plate collector and the single slope solar still with flat plate collector. The |

detail discussion of the efficiency is highlighted in lines 365-373.

The solar still in the present study is made with locally sourced materials and as at the time of construction the average cost is approximately $ 140 including the flat plate collector. Without the flat plate collector the average cost is $ 40. The analysis for the cost per liter of distilled water based on Kabeel et al. [73] has been included to compare the cost of distilled water production per litre. The detail analysis is highlighted in lines 387-408. Furthermore, it is observed that the present design is better than some of the earlier designs based on cost of production per litre of water.

**AUTHOR'S RESPONSE TO REFEREE'S COMMENTS (RC1: )**

| Referee's Comment | Author's Response: Relevant lines are highlighted in torquoiso color |
|---|---|
| S10: Rephrase this line (havoc is not the appropriate terminology). For instance: The major problems caused by scarcity of drinking water | Rephrased as highlighted in line 10-12 |
| S20: TDS-measurement is in this case probably measured with electrical conductivity. The meter is measuring the EC and a linear relation is assumed between EC and TDS. So on the display there is a TDS-reading. This linear relation is not very accurate. The proper way to measure TDS is to measure all the ions in the water matrix in mg/l and add them up to obtain the total amount of dissolved ions. So I would suggest only to talk about EC in this paper. | A digital meter that measures both the TDS and the EC was used. The modes were only interchanged to capture the required parameter at each specific time. As reported in lines 224-227. The total dissolved solid in the water samples was measured using a digital conductivity meter by Mettler Toledo with ±0.5 % conductivity accuracy. However, emphasis has been reduced on TDS due to the suggestion made by the reviewer (RC1). |

| | |
|---|---|
| S51: Here electrolysis is mentioned but this is a process to split water in $H_2$ and $O_2$. What is meant is electrodialysis. This is a process with two different membranes and two electrodes pulling ions trough the membranes resulting in one diluate (less ions) and one concentrate (more ions) stream. | The suggestions has been effected as suggested by the reviewer and highlighted in lines 51-55. |
| S90: Why is roof water carcinogenic? I can imagine that it contains bacteria and viruses (from the birds on the roof) and it can contain metals like iron and zinc from the roof material but I cannot imagine that it contains carcinogenic compounds. If this is possible please refer to literature to prove this. | This observation has been reported in the literature as cited in line 75-77. |
| S96: Rephrase "as a result of indiscriminate drinking of water". Drinking water according to the WHO-guidelines will never contain water borne diseases. So probably you mean that people use water that is not treated to drinking water or the quality is not meeting the WHO-guidelines. | This has been effected as highlighted in lines 94-96. |
| S106: Mention here also the $m^2$ of the solar collector because this surface area contributes to the solar heat that is collected during the experiment. | The reviewer pointed out the need to mention the area of the solar collector that received the heat from the sun. This area is 1 square meter as described and highlighted in lines 158 - 161. |
| S207: Avoid terminology like "ridiculous" | It has been corrected |
| S210: Mention the brand of the measuring equipment (but the TDS meter is in fact a conductivity meter) | The brand of the measuring equipment is a digital conductivity meter by Mettler Toledo with $\pm0.5$ % conductivity accuracy. The digital meter was used to measure both the TDS and the EC. This is described in lines 224 - 227. |
| S281: Mention which graph is the active and which graph is the passive setup (probably: a and d are active and b and c are passive setups) | Figures 4 (a and d) gives the distillate yield for the active solar still while Figures 4(b and c) represent the distillate yield for the passive still. This is captured as highlighted in lines 298-300. |
| S323: Mention that EC and TDS-removal rate is not very relevant in this case because the starting TDS is already below | We appreciate the suggestions of the reviewer and this line has been |

| | |
|---|---|
| the WHO-guidelines. If seawater or brackish water was investigated this was a more relevant parameter. And for seawater the reduction rate should be something like 99.9% to obtain drinking water. | modified as highlighted in lines 341-346. |
| S336: The table here (label has no number!!!) shows a unit I cannot understand: Maximum daily production rate (kg/m2hr). So probably the proper unit is kg/(m2.day) In the table you should mention for comparison your results for the passive setup and the active setup. And mention if the m2 of the solar collector is used in this calculation. Because in fact you should refer the production to the total m2 surface area you use to collect solar heat. | The tables here have been numbered and the data are presented in SI units. Daily production rate is now written in litres per square meter of solar collector per day. This explanation addressed the observations raised by the reviewer. |
| S346 The XXXX should be replaced by numbers??? Please do so. Otherwise delete this part. | The XXXX has been replaced by numbers as suggested by the reviewer. Other relevant information has been added to improve the quality of the paper. This is captured in lines 360 -368 as highlighted in the paper. |

**AUTHOR'S RESPONSE TO REFEREE'S COMMENTS (RC2)**

| Referee's Comment | Author's Response: **Relevant lines are highlighted in green color** |
|---|---|
| The removal of TDS and EC was studied using a locally made solar distillation installation. Various water sources were used and compared to other experiments described in literature. The paper is reasonably well written, but lacks a clear objective and a good discussion of the results with literature | These general observations have been carefully rectified based on reviewer's comment as follows. |

| | |
|---|---|
| **General comments**

A clear objective at the end of the introduction is missing. How does this relate to previous research in the area? What is novel? Only location is not sufficient.. Is the design novel? - It should be explained why solar stills are used to treat the water mentioned water sources. Probably there are more cost effective ways to treat groundwater, rain water and surface water. - EC and TDS is not sufficient to judge the treatment performance since these are not indicators for microbial contamination e.g. - Comparing EC and TDS to WHO guidelines is not sufficient to judge performance. - The discussion with literature should be included in the sections describing the results (now they are separated). - Cost analyses should be made in comparison with the production, so xxx Cm3 - Language, including tenses, should be checked - Redundant information should be deleted. - Avoid too general introduction | A clear objective has been written close to the end of the introduction as highlighted in lines 84-88. The relationship between the present study and earlier works are established by comparing design, performance, efficiency and cost as highlighted in lines 111-114, 295-301, 352-355, 362-363 and 402-405.

The reviewer raises issues on TDS and EC measurement since we did not check the microbial level or activities. Many papers discussed TDS and EC without specific emphasis on microbial level. The scope of the study does not consider the level of microbial contamination in the water sample before and after the desalination. TDS and EC tested before and after desalination are just in addition to the effect of solar insolation and temperature variations on the yield of the distillate from the constructed solar still. These were carried out to judge the performance of the constructed solar still. Other yardstick/parameter exist but not within the scope of this study. The main objective is to evaluate the performance of the Solar still based on the obtained yield, WHO standard on the TDS and EC of the output, Cost reduction (based on the locally sourced materials used in construction), etc. TDS and EC measurement are one of the ways by which Solar still performance is checked in the literature. Future work may include checking the level of microbial contamination before and after desalination |
| Line 21-22 and 23-26 | Correction made based on reviewer's comment |
| Line 28-38 | Correction made based on reviewer's comment |
| Line 39 | Effected |
| Line 46 | Location specified as observed |
| Line 48-51 | Corrected as suggested |

| | |
|---|---|
| Line 54-57 | Corrected as suggested |
| Line 81, 84 & 89 | Word has been replaced and explanation on how this work solves the water purification problem was given to reflect the authors' opinion. |
| Line 95-96 | The statement has been rephrased as suggested and new statement is not bold or too assertive |
| Line 138-145 | Correction implemented as suggested by the reviewer, relevant references are included. |
| Line 162-167 | The statement has been rephrased as suggested and new statement is not bold or too assertive |
| Line 174 | The overview of all the experimental settings is given in lines 162 – 187 and 190-222. Figures 1 and 2 have shown the experimental set-up, the detail overview is not considered necessary in the author's opinion. Other issues raised regarding duplicates in experiments and water sampling have been captured under experimental design |
| Line 175 | Performance evaluation is now put under material and methods as suggested by the reviewer |
| Line 183-186: Explain what design variables were varied and evaluated for optimized performance | None of the design variables mentioned in the session were varied or evaluated for optimization. All will do was that we compared the performance of passive flat plate collector against the active type. |
| Line 196-209: should be rephrased (or deleted) based on the general comments above | We have checked this; there is no reason to rephrase or delete. It is important to the article in our own opinion |
| Line 210: dissolved solids are not "particles"; What is a "digital TDS meter", (type/measurement method, etc.)? | Highlighted in lines 224-227 |
| Line 219-226:………….should be more extensive and part of Materials and Methods section | ………..This has been elaborated in the material and methods section |
| Line 228-230: consider deleting | No need for deletion but modified |
| Line 232: why randomly selected days? Is there another way to present | Experiments were carried out on several days. But we cannot present all the |

| all days? | results because of space. And the results equally behave the same in as much as the solar radiation for the days under consideration look similar and it is the same experimental condition and water sample. In some case some experiments were even repeated. So, the 9 days selected are 4 days for active solar still and 5 days for the passive type. |
|---|---|
| Line 236-237: is this relationship known from literature, then discuss this with literature | These have been discussed as highlighted in lines 273-284 |
| Line 238 - 243 | Corrected based on reviewer's suggestions |
| Line 256 and onwards: how does it compare with other studies? | References given as highlighted in lines 295-301 |
| Line 267-269: not new

Line 274: results = resulted

Line 284: has = had | These are editorial errors and have been corrected |
| Line 275: do not use "significantly" when statistical analyses are not performed | This has been rectified based on reviewer's suggestion |
| Line 294-307: can be deleted, because the graphs represent the same data of previous graphs and do not give extra information. | These graphs cannot be deleted because the parameters considered are different even though they look similar. The authors considered the graphs necessary and thereby retained them (Figures 3, 4 & 5) |

---

## Author Comment (AC4) · 13 Sep 2020

Reviewers Comment: The removal of TDS and EC was studied using a locally made solar distillation installation. Various water sources were used and compared to other experiments described in literature. The paper is reasonably well written, but lacks a clear objective and a good discussion of the results with literature.

Author s Response: These general observations have been carefully rectified based on reviewer's comment as follows.

General comments by Reviewer: A clear objective at the end of the introduction is

[Figure]

missing. How does this relate to previous research in the area? What is novel? Only location is not sufficient.. Is the design novel? - It should be explained why solar stills are used to treat the water mentioned water sources. Probably there are more cost effective ways to treat groundwater, rain water and surface water. - EC and TDS is not sufficient to judge the treatment performance since these are not indicators for microbial contamination e.g. - Comparing EC and TDS to WHO guidelines is not sufficient to judge performance. - The discussion with literature should be included in the sections describing the results (now they are separated). - Cost analyses should be made in comparison with the production, so xxx Cm3 - Language, including tenses, should be checked - Redundant information should be deleted. - Avoid too general introduction

Author s Response: A clear objective has been written close to the end of the introduction as highlighted in lines 84-88. The relationship between the present study and earlier works are established by comparing design, performance, efficiency and cost as highlighted in lines 111-114, 295-301, 352-355, 362-363 and 402-405. The reviewer raises issues on TDS and EC measurement since we did not check the microbial level or activities. Many papers discussed TDS and EC without specific emphasis on microbial level. The scope of the study does not consider the level of microbial contamination in the water sample before and after the desalination. TDS and EC tested before and after desalination are just in addition to the effect of solar insolation and temperature variations on the yield of the distillate from the constructed solar still. These were carried out to judge the performance of the constructed solar still. Other yardstick/parameter exist but not within the scope of this study. The main objective is to evaluate the performance of the Solar still based on the obtained yield, WHO standard on the TDS and EC of the output, Cost reduction (based on the locally sourced materials used in construction), etc. TDS and EC measurement are one of the ways by which Solar still performance is checked in the literature. Future work may include checking the level of microbial contamination before and after desalination.

Reviewer's Comment Line 21-22 and 23-26 Author's Response: Correction made

based on reviewer's comment

Reviewer's Comment Line 28-38 Author s Response: Correction made based on reviewer's comment

Reviewer's Comment Line 39 Author s Response: Effected based on reviewer's comment

Reviewer's Comment Line 46 Author s Response: Location specified as observed by the reviewer

Reviewer's Comment Line 48-51 Author s Response: Corrected as suggested by the reviewer

Reviewer's Comment Line 54-57 Author s Response: Corrected as suggested by the reviewer

Reviewer's Comment Line 81, 84 & 89 Author s Response: Word has been replaced and explanation on how this work solves the water purification problem was given to reflect the authors' opinion.

Reviewer's Comment Line 95-96 Author s Response: The statement has been rephrased as suggested and new statement is not bold or too assertive

Reviewer's Comment Line 138-145 Author s Response: Correction implemented as suggested by the reviewer, relevant references are included.

Reviewer's Comment Line 162-167 Author s Response: The statement has been rephrased as suggested and new statement is not bold or too assertive.

Reviewer's Comment Line 174 Author s Response: The overview of all the experimental settings is given in lines 162 – 187 and 190-222. Figures 1 and 2 have shown the experimental set-up, the detail overview is not considered necessary in the author's opinion. Other issues raised regarding duplicates in experiments and water sampling have been captured under experimental design

Reviewer's Comment Line 175 Author s Response: Performance evaluation is now put under material and methods as suggested by the reviewer

Reviewer's Comment Line 183-186 Explain what design variables were varied and evaluated for optimized performance

Author s Response: None of the design variables mentioned in the session were varied or evaluated for optimization. All we do was that we compared the performance of passive flat plate collector against the active type.

Reviewer's Comment Line 196-209: should be rephrased (or deleted) based on the general comments above Author s Response: We have checked this; there is no reason to rephrase or delete. It is important to the article in our own opinion

Reviewer's Comment Line 210: dissolved solids are not "particles"; What is a "digital TDS meter", (type/measurement method, etc.)? Author s Response: Corrected as Highlighted in lines 224-227

Reviewer's Comment Line 219-226:…………..should be more extensive and part of Materials and Methods section Author s Response: This has been elaborated in the material and methods section

Reviewer's Comment Line 228-230: consider deleting Author s Response: No need for deletion but modified

Reviewer's Comment Line 232: why randomly selected days? Is there another way to present all days? Author s Response: Experiments were carried out on several days. But we cannot present all the results because of space. And the results equally behave the same in as much as the solar radiation for the days under consideration look similar and it is the same experimental condition and water sample. In some case some experiments were even repeated. So, the 9 days selected are 4 days for active solar still and 5 days for the passive type.

Reviewer's Comment Line 236-237: is this relationship known from literature, then

discuss this with literature. Author s Response: These have been discussed as highlighted in lines 273-284

Reviewer's Comment Line 238 - 243 Author s Response: Corrected based on reviewer's suggestions

Reviewer's Comment Line 256 and onwards: how does it compare with other studies? Author s Response: References given as highlighted in lines 295-301

Reviewer's Comment Line 267-269, 274, 284. Author s Response: Editorial errors and have been rectified as suggested by the reviewer

Reviewer's Comment Line 275: Do not use "significantly" when statistical analyses are not performed Author s Response: This has been rectified based on reviewer's suggestion

Reviewer's Comment Line 294-307: can be deleted, because the graphs represent the same data of previous graphs and do not give extra information. Author s Response: These graphs cannot be deleted because the parameters considered are different even though they look similar. The authors considered the graphs necessary and thereby retained them (Figures 3, 4 & 5)

Please also note the supplement to this comment:
https://dwes.copernicus.org/preprints/dwes-2020-5/dwes-2020-5-AC4-supplement.pdf

**Supplement:**

| TOPICAL EDITOR'S COMMENTS | AUTHOR'S RESPONSE: Relevant lines are Highlighted in yellow colour in the main article |
| --- | --- |
| **Why did you choose to distill water with already low salt contents? (Other treatment systems are more efficient then), better to focus on seawater.** | Apart from the coastal region of Nigeria where people are forced by circumstance to process salty water for domestic use, the commonly available water in some rural areas is not pure due to dissolved organic and inorganic materials. In some locations (e.g Ile-ife Osun state: 7.4905°N, 4.5521°E) ), the economic power of the indigenes is very low and living standard is very poor. Therefore, the present work aimed at providing indigenous distillation method for locals in order to make drinkable water available at low cost. Justification is highlighted in lines 78-84. |
| **What is new in relation to the already existing solar distillators?** | Although solar distillation is not a new technology, likewise the method/structure of solar still (that is single slope conventional type) adopted in this research. However, the experimental design and the setup are location specific. The tilt angle of the glass condenser which significantly affects the output of the solar still are chosen based on the latitude of the research location, in this case 7.5175° N. Hence, the glass cover was kept at 17°52'', (i.e. the (7.5175° N) plus 10°)…….Highlighted in lines 106-109 |
| **Explain in materials and methods section your experimental design, what do you mean with "dirty water", and how the analyses were done (instruments)?** | **"Dirty water"** in this context is referred to as the water collected from stagnant water most often where passer bye (people) urinates, defecate and deposit refuse. It is heavily polluted with algae, spirogyra and refuse/dirt of all kinds. |

| | |
|---|---|
| | **"Experimental Design"** |
| | Two sets of experiments were prepared: the conventional solar still (CSS) and conventional solar still with a flat plate collector (CSS-FPC). In this experimental work, the conventional solar still was fabricated with a square stainless-steel sheet of 1 $m^2$ and 2 mm thickness. Figure 1 shows the isometric and the exploded view of the experimental setup, while Figure 2 shows the photograph of the experimental setup. The detail experimental design has been carefully explained as highlighted in lines 189-222. Analysis were carried out by first obtaining data using relevant measuring instruments and the results were analyzed in form of evaluation and comparison of results with earlier works in the literature. Important evaluation of the present work is highlighted in lines 224-252. |
| **Discuss your results in relation to literature (is it in line, is it better and why..)** | Several authors have worked on performance evaluation of solar still of different configurations. Their results are hereby compared with that of the present studies as explained and highlighted in lines 276-287, and lines 298-304. This is with the understanding that the performance of any solar still is dependent on the following factors: location under consideration (inherent/current climatic and atmospheric condition); diurnal irradiance and other specified experimental conditions.

Direct comparison of the performance of the solar still of present work with the available results in the literature is shown in Tables 3 and 4. Comparing TDS and EC, the present work is close to the |

| | earlier works in terms of TDS and EC reduction for heavily polluted water and freshly dug water which is a major problem facing the rural communities around the location where the experiments were conducted.

Other comparisons are highlighted in lines 356- 361. |
|---|---|
| **Discuss with literature the "efficiency" of the system (=efficacy and costs) in relation to other systems.** | Different methods exist for water purification using solar desalination system. The TDS and the electrical conductivity of the produced desalinated water from the four different sources have been compared with some results available in the literature using different configurations of solar desalination system. The maximum daily yield of the present work is better than most of the existing solar still as shown in Table 4.

The average of the overall daily efficiencies of the conventional solar still without flat plate collector and the single slope solar still with flat plate collector are 13.906 % and 16.298 % respectively. This shows an improvement of 14.67 % with the inclusion of the flat plate with the conventional type. Since these values are dependent on the weather, climate and the atmospheric conditions with the diurnal irradiance coupled with the still design, hence it is difficult to compare with existing designs in the literature.

The daily production efficiency, $\epsilon_d$ of the still are 15.85 % and 26.25 % respectively for the conventional solar still without flat plate collector and the single slope solar still with flat plate collector. The |

detail discussion of the efficiency is highlighted in lines 365-373.

The solar still in the present study is made with locally sourced materials and as at the time of construction the average cost is approximately $ 140 including the flat plate collector. Without the flat plate collector the average cost is $ 40. The analysis for the cost per liter of distilled water based on Kabeel et al. [73] has been included to compare the cost of distilled water production per litre. The detail analysis is highlighted in lines 387-408. Furthermore, it is observed that the present design is better than some of the earlier designs based on cost of production per litre of water.

**AUTHOR'S RESPONSE TO REFEREE'S COMMENTS (RC1: )**

| Referee's Comment | Author's Response: Relevant lines are highlighted in torquoiso color |
|---|---|
| S10: Rephrase this line (havoc is not the appropriate terminology). For instance: The major problems caused by scarcity of drinking water | Rephrased as highlighted in line 10-12 |
| S20: TDS-measurement is in this case probably measured with electrical conductivity. The meter is measuring the EC and a linear relation is assumed between EC and TDS. So on the display there is a TDS-reading. This linear relation is not very accurate. The proper way to measure TDS is to measure all the ions in the water matrix in mg/l and add them up to obtain the total amount of dissolved ions. So I would suggest only to talk about EC in this paper. | A digital meter that measures both the TDS and the EC was used. The modes were only interchanged to capture the required parameter at each specific time. As reported in lines 224-227. The total dissolved solid in the water samples was measured using a digital conductivity meter by Mettler Toledo with ±0.5 % conductivity accuracy. However, emphasis has been reduced on TDS due to the suggestion made by the reviewer (RC1). |

| | |
|---|---|
| S51: Here electrolysis is mentioned but this is a process to split water in $H_2$ and $O_2$. What is meant is electrodialysis. This is a process with two different membranes and two electrodes pulling ions trough the membranes resulting in one diluate (less ions) and one concentrate (more ions) stream. | The suggestions has been effected as suggested by the reviewer and highlighted in lines 51-55. |
| S90: Why is roof water carcinogenic? I can imagine that it contains bacteria and viruses (from the birds on the roof) and it can contain metals like iron and zinc from the roof material but I cannot imagine that it contains carcinogenic compounds. If this is possible please refer to literature to prove this. | This observation has been reported in the literature as cited in line 75-77. |
| S96: Rephrase "as a result of indiscriminate drinking of water". Drinking water according to the WHO-guidelines will never contain water borne diseases. So probably you mean that people use water that is not treated to drinking water or the quality is not meeting the WHO-guidelines. | This has been effected as highlighted in lines 94-96. |
| S106: Mention here also the $m^2$ of the solar collector because this surface area contributes to the solar heat that is collected during the experiment. | The reviewer pointed out the need to mention the area of the solar collector that received the heat from the sun. This area is 1 square meter as described and highlighted in lines 158 - 161. |
| S207: Avoid terminology like "ridiculous" | It has been corrected |
| S210: Mention the brand of the measuring equipment (but the TDS meter is in fact a conductivity meter) | The brand of the measuring equipment is a digital conductivity meter by Mettler Toledo with $\pm0.5$ % conductivity accuracy. The digital meter was used to measure both the TDS and the EC. This is described in lines 224 - 227. |
| S281: Mention which graph is the active and which graph is the passive setup (probably: a and d are active and b and c are passive setups) | Figures 4 (a and d) gives the distillate yield for the active solar still while Figures 4(b and c) represent the distillate yield for the passive still. This is captured as highlighted in lines 298-300. |
| S323: Mention that EC and TDS-removal rate is not very relevant in this case because the starting TDS is already below | We appreciate the suggestions of the reviewer and this line has been |

| | |
|---|---|
| the WHO-guidelines. If seawater or brackish water was investigated this was a more relevant parameter. And for seawater the reduction rate should be something like 99.9% to obtain drinking water. | modified as highlighted in lines 341-346. |
| S336: The table here (label has no number!!!) shows a unit I cannot understand: Maximum daily production rate (kg/m2hr). So probably the proper unit is kg/(m2.day) In the table you should mention for comparison your results for the passive setup and the active setup. And mention if the m2 of the solar collector is used in this calculation. Because in fact you should refer the production to the total m2 surface area you use to collect solar heat. | The tables here have been numbered and the data are presented in SI units. Daily production rate is now written in litres per square meter of solar collector per day. This explanation addressed the observations raised by the reviewer. |
| S346 The XXXX should be replaced by numbers??? Please do so. Otherwise delete this part. | The XXXX has been replaced by numbers as suggested by the reviewer. Other relevant information has been added to improve the quality of the paper. This is captured in lines 360 -368 as highlighted in the paper. |

**AUTHOR'S RESPONSE TO REFEREE'S COMMENTS (RC2)**

| Referee's Comment | Author's Response: **Relevant lines are highlighted in green color** |
|---|---|
| The removal of TDS and EC was studied using a locally made solar distillation installation. Various water sources were used and compared to other experiments described in literature. The paper is reasonably well written, but lacks a clear objective and a good discussion of the results with literature | These general observations have been carefully rectified based on reviewer's comment as follows. |

| | |
|---|---|
| **General comments**

A clear objective at the end of the introduction is missing. How does this relate to previous research in the area? What is novel? Only location is not sufficient.. Is the design novel? - It should be explained why solar stills are used to treat the water mentioned water sources. Probably there are more cost effective ways to treat groundwater, rain water and surface water. - EC and TDS is not sufficient to judge the treatment performance since these are not indicators for microbial contamination e.g. - Comparing EC and TDS to WHO guidelines is not sufficient to judge performance. - The discussion with literature should be included in the sections describing the results (now they are separated). - Cost analyses should be made in comparison with the production, so xxx Cm3 - Language, including tenses, should be checked - Redundant information should be deleted. - Avoid too general introduction | A clear objective has been written close to the end of the introduction as highlighted in lines 84-88. The relationship between the present study and earlier works are established by comparing design, performance, efficiency and cost as highlighted in lines 111-114, 295-301, 352-355, 362-363 and 402-405.

The reviewer raises issues on TDS and EC measurement since we did not check the microbial level or activities. Many papers discussed TDS and EC without specific emphasis on microbial level. The scope of the study does not consider the level of microbial contamination in the water sample before and after the desalination. TDS and EC tested before and after desalination are just in addition to the effect of solar insolation and temperature variations on the yield of the distillate from the constructed solar still. These were carried out to judge the performance of the constructed solar still. Other yardstick/parameter exist but not within the scope of this study. The main objective is to evaluate the performance of the Solar still based on the obtained yield, WHO standard on the TDS and EC of the output, Cost reduction (based on the locally sourced materials used in construction), etc. TDS and EC measurement are one of the ways by which Solar still performance is checked in the literature. Future work may include checking the level of microbial contamination before and after desalination |
| Line 21-22 and 23-26 | Correction made based on reviewer's comment |
| Line 28-38 | Correction made based on reviewer's comment |
| Line 39 | Effected |
| Line 46 | Location specified as observed |
| Line 48-51 | Corrected as suggested |

| | |
|---|---|
| Line 54-57 | Corrected as suggested |
| Line 81, 84 & 89 | Word has been replaced and explanation on how this work solves the water purification problem was given to reflect the authors' opinion. |
| Line 95-96 | The statement has been rephrased as suggested and new statement is not bold or too assertive |
| Line 138-145 | Correction implemented as suggested by the reviewer, relevant references are included. |
| Line 162-167 | The statement has been rephrased as suggested and new statement is not bold or too assertive |
| Line 174 | The overview of all the experimental settings is given in lines 162 – 187 and 190-222. Figures 1 and 2 have shown the experimental set-up, the detail overview is not considered necessary in the author's opinion. Other issues raised regarding duplicates in experiments and water sampling have been captured under experimental design |
| Line 175 | Performance evaluation is now put under material and methods as suggested by the reviewer |
| Line 183-186: Explain what design variables were varied and evaluated for optimized performance | None of the design variables mentioned in the session were varied or evaluated for optimization. All will do was that we compared the performance of passive flat plate collector against the active type. |
| Line 196-209: should be rephrased (or deleted) based on the general comments above | We have checked this; there is no reason to rephrase or delete. It is important to the article in our own opinion |
| Line 210: dissolved solids are not "particles"; What is a "digital TDS meter", (type/measurement method, etc.)? | Highlighted in lines 224-227 |
| Line 219-226:………….should be more extensive and part of Materials and Methods section | ………..This has been elaborated in the material and methods section |
| Line 228-230: consider deleting | No need for deletion but modified |
| Line 232: why randomly selected days? Is there another way to present | Experiments were carried out on several days. But we cannot present all the |

| all days? | results because of space. And the results equally behave the same in as much as the solar radiation for the days under consideration look similar and it is the same experimental condition and water sample. In some case some experiments were even repeated. So, the 9 days selected are 4 days for active solar still and 5 days for the passive type. |
|---|---|
| Line 236-237: is this relationship known from literature, then discuss this with literature | These have been discussed as highlighted in lines 273-284 |
| Line 238 - 243 | Corrected based on reviewer's suggestions |
| Line 256 and onwards: how does it compare with other studies? | References given as highlighted in lines 295-301 |
| Line 267-269: not new

Line 274: results = resulted

Line 284: has = had | These are editorial errors and have been corrected |
| Line 275: do not use "significantly" when statistical analyses are not performed | This has been rectified based on reviewer's suggestion |
| Line 294-307: can be deleted, because the graphs represent the same data of previous graphs and do not give extra information. | These graphs cannot be deleted because the parameters considered are different even though they look similar. The authors considered the graphs necessary and thereby retained them (Figures 3, 4 & 5) |

---

## Author Response (AR2)

**Topical Editor Decision: Publish subject to minor revisions (review by editor) (29 Nov 2020) by Luuk Rietveld**

**Comments to the Author:** Several comments are well addressed. However, before publication the following should still be revised:

**Query 1**
Avoid too general introductions. So, e.g. delete sentences on lines 25-29; 31-41; 115-118; 119-123; 126-130; 133-135; 232-234;
**Response 1**
All identified lines for deletion have been deleted and some sentences were modified to preserve the flow of information.

**Query 2**
Merge chapters 1 & 2
**Response 2**
Chapters 1 and 2 merged.

**Query 3**
As commented by reviewer delete references to TDS in entire document, since the measurement is indirect by EC meter. So only discuss your results based on EC removal.
**Response 3**
The references to TDS in the entire document have been deleted and the discussions were based on EC.

**Query** 4
When introducing abbreviations, do it when word appears for the first time and then ALWAYS use abbreviation, such as EC, CSS (do not use SS)..
**Response 4**
The issue on abbreviation has been corrected in the entire document

**Query 5**
Line 43 (and other parts of the manuscript): specify what is meant by "heavily polluted" water (surface water, pond water, drainage water?)
**Response 5**
Heavily polluted is described as stagnant surface water that retain dirt because it doesn't flow.

**Query 6**
Line 55: CSS has environmental pollution, since a brine is produced, so this statement is incorrect.
**Response 6**
The sentence has been modified as highlighted in line 46 of the revised manuscript.

**Query 7**
Line 65-67: repetition, avoid redundancies
**Response 7**
The repetition has been addressed

**Query 8**
Line 69-71: as suggested by reviewer, it is highly unlikely that roof top water with rusted iron is carcinogenic, so adjust
**Response 8**
The statement has been adjusted as highlighted in line 32-33.

**Query 9**
Avoid words like "very" (line 75) and "huge" (line 366)
**Response 9**
These words have been replaced or removed

**Query 10**
Line 96-98: rephrase sentence
**Response 10**
The phrase is modified as shown in lines 63-65 of the revised manuscript.

**Query 11**
Line 236 and onwards: described the results in the past tense
**Response 11**
The aspect of discussion of result from lines 236 has been modified to reflect the use of past tense where necessary.

**Query 12**
Line 303-313: merge two section
**Response 12**
The two sections in Lines 303-313 have been merged and information about TDS has been deleted in response to the earlier suggestion by the reviewer. This is shown in lines 266-272 of the revised manuscript.

**Query 13**
Line 314-320: integrate the discussion with previous research with the other results section (no specific discussion section).
**Response 13**
The most important results that need to be compared is performance evaluation of the solar stills. This comparison is discussed in lines 273 -279 and tabulated in table 2, the results discussed in other sections are specific parameter of the solar still, these are better discussed separately for clarity.

**Query 14**
Line 337-354: too extensive explanation. Make it more concise and base it on literature.
**Response 14**

The analysis of cost of distilled water per litre based on the literature is explained concisely as shown in lines 296-302.

**Query 15**
Line 355: compare = comparing
**Response 15**
This has been corrected as shown in line 303

**Query 16**
Delete Table 1 & Table 2 and integrate data in text
**Response 16**
Table 1 has been deleted but Table 2 has been modified to exclude TDS results and some data have been integrated in the text as suggested.

**Query 17**
Delete Table 4, since the information is too specific
**Response 17**
Table 4 shows important information on comparison of past works with the present research. The authors feel this table is very relevant and should be retained but with new title "Table 2". Moreover, this was requested by one of the reviewers in the first round of reviews. Thank you.